# *DALRD3* encodes a protein mutated in epileptic encephalopathy that targets arginine tRNAs for 3-methylcytosine modification

Jenna M. Lentini[1], Hessa S. Alsaif[2], Eissa Faqeih[3], Fowzan S. Alkuraya [2,4] & Dragony Fu [1✉]

In mammals, a subset of arginine tRNA isoacceptors are methylated in the anticodon loop by the METTL2 methyltransferase to form the 3-methylcytosine (m3C) modification. However, the mechanism by which METTL2 identifies specific tRNA arginine species for m3C formation as well as the biological role of m3C in mammals is unknown. Here, we show that human METTL2 forms a complex with DALR anticodon binding domain containing 3 (DALRD3) protein to recognize particular arginine tRNAs destined for m3C modification. DALRD3-deficient human cells exhibit nearly complete loss of the m3C modification in tRNA-Arg species. Notably, we identify a homozygous nonsense mutation in the *DALRD3* gene that impairs m3C formation in human patients exhibiting developmental delay and early-onset epileptic encephalopathy. These findings uncover an unexpected function for the DALRD3 protein in the targeting of distinct arginine tRNAs for m3C modification and suggest a crucial biological role for DALRD3-dependent tRNA modification in proper neurological development.

[1] Department of Biology, Center for RNA Biology, University of Rochester, Rochester, NY, USA. [2] Department of Genetics, King Faisal Specialist Hospital and Research Center, Riyadh, Saudi Arabia. [3] Section of Medical Genetics, Children's Specialist Hospital, King Fahad Medical City, Riyadh, Saudi Arabia. [4] Department of Anatomy and Cell Biology, College of Medicine, Alfaisal University, Riyadh, Saudi Arabia. ✉email: dragonyfu@rochester.edu

The proper maturation and function of tRNAs has emerged as a critical modulator of biological processes ranging from gene regulation to development[1–3]. In particular, tRNAs are subject to a diverse range of chemical modifications that play major roles in their folding, stability and function[4–6]. The critical role of tRNA modification in organismal physiology and fitness is highlighted by the numerous human diseases that have been associated with defects in tRNA modification including neurological disorders, mitochondrial pathologies, and cancer (reviewed in refs. [7–12]).

The 3-methylcytosine (m3C) modification at position 32 of the anticodon loop in certain tRNAs is a eukaryotic-specific modification conserved from yeast to mammals. The m3C modification is predicted to play a role in anticodon folding and function since the nucleotide at residue 32 forms a non-canonical interaction with residue 38 to maintain the conformation of the anticodon loop[13–15]. In the yeasts *Saccharomyces cerevisiae* and *Schizosaccharomyces pombe*, the m3C modification is present in tRNA-Thr and tRNA-Ser isoacceptors[16–18]. Mammals also possess m3C in all tRNA-Ser and tRNA-Thr isoacceptors but have also evolved to harbor m3C in mitochondrial-encoded tRNA-Ser-UGA and tRNA-Thr-UGU as well as cytoplasmic tRNA-Arg-UCU and CCU[17–22].

In *S. cerevisiae*, the Trm140p methyltransferase is responsible for m3C formation in tRNA-Ser and Thr isoacceptors[23,24]. Interestingly, the fission yeast *S. pombe* expresses two Trm140 homologs encoded by the *Trm140* and *Trm141* genes that are separately responsible for catalyzing m3C in tRNA-Thr and tRNA-Ser, respectively[17]. In *S. cerevisiae*, Trm140p recognizes tRNA-Thr substrates via a sequence element encompassing nucleotides 35–37 of the anticodon loop that also includes the t6A modification at position 37[25]. The recognition of tRNA-Ser isoacceptors by yeast Trm140 homologs is also dependent upon modification of position 37, which can be either t6A or i6A depending on the tRNA-Ser isoacceptor[17,25]. Notably however, *S. cerevisiae* Trm140p also requires an interaction with seryl-tRNA synthetase in order to methylate the corresponding seryl-tRNA[25]. This utilization of the seryl-tRNA synthetase ensures the proper catalysis of m3C on all tRNA serine isotypes since the seryl-tRNA synthetase has evolved to recognize the unusually long variable loop and diverse tertiary structure elements present in the various tRNA-Ser species[26,27]. Recent studies in *Trypanosoma brucei* have also uncovered an unusual mechanism by which TRM140 interacts with the ADAT2/3 deaminase complex to form m3C at position 32 as a pre-requisite for subsequent deamination by the ADAT2/3 enzyme to form 3-methyluridine[28]. Collectively, these studies highlight the complex circuitry of modifications in the anticodon loop (reviewed in refs. [29,30]).

Four human homologs of Trm140p have been identified by sequence homology, which are encoded by the *METTL2A, METTL2B, METTL6,* and *METTL8* genes[17,31]. METTL6 is responsible for the catalysis of m3C at position 32 in tRNA-Ser while METTL8 has been proposed to play a role in mRNA modification[32]. *METTL2A* and *METTL2B* encode paralogous proteins that share 99% amino acid sequence identity and form their own vertebrate-specific phylogenetic clade with METTL8 homologs within the Trm140p homology tree[17,23,24]. Like *S. cerevisiae* Trm140p and *S. pombe* Trm140p, METTL2A and 2B are required for the formation of m3C in threonyl-tRNAs[32]. In addition, human METTL2A and METTL2B are required for m3C formation in the arginine tRNA isoacceptors, tRNA-Arg-CCU and Arg-UCU. The presence of m3C in tRNA-Arg-CCU and Arg-UCU has been detected in multiple mammalian species but not in yeast or plants[17,19,20,33,34], suggesting that METTL2A/B-catalyzed modification of arginine tRNAs evolved within the animal kingdom. However, the molecular mechanism by which METTL2A and 2B recognizes only a subset of arginine tRNA substrates as well as the biological roles of m3C modification in mammals are unknown.

Here, we demonstrate that METTL2A and 2B interact with DALRD3, a previously uncharacterized protein harboring a putative anticodon-binding domain found in arginyl tRNA synthetases. Using gene editing, we show that loss of DARLD3 expression in human cells abolishes m3C formation in arginine tRNAs that can be rescued with re-expression of full-length DALRD3. Strikingly, we find that DALRD3 is mutated in human patients exhibiting a severe form of epileptic encephalopathy and that these patients no longer exhibit m3C modification in their arginine tRNAs. Altogether, this study uncovers an unanticipated role for the DALRD3 protein in the recognition of specific arginine tRNAs for METTL2-catalyzed m3C formation and implicates the m3C modification in proper neurological function.

## Results

**METTL2 interacts with a putative tRNA-binding protein DALRD3.** To identify proteins that interact with human METTL2A or 2B, we generated 293T human embryonic cell lines stably expressing METTL2A or METTL2B fused to the Twin-Strep purification tag[35]. As a comparison, we also generated a 293T cell line stably expressing Strep-tagged METTL6, a different human Trm140 homolog. Strep-METTL2A, METTL2B and METTL6 were affinity purified from whole cell extracts on streptactin resin, eluted with biotin and analyzed by silver staining to detect interacting proteins. We identified a closely-migrating doublet of bands at ~50 kDa along with a band at ~30 kDa that were enriched with METTL2A and METTL2B compared to the control purification (Fig. 1a, arrowheads). Immunoblotting with anti-Strep tag antibodies revealed that one of the 50 kDa bands was purified Strep-METTL2A/B (Supplementary Fig. 1). The METTL6 purification yielded bands at ~60, 40, and 30 kDa (Fig. 1a, arrowheads). The 40 kDa band corresponds to purified Strep-METTL6 as detected by immunoblotting with anti-Strep tag antibodies (Supplementary Fig. 1).

To identify METTL-interacting proteins, the total eluates from each purification were processed for peptide identification by liquid chromatography-tandem mass spectrometry (LC-MS). As expected, peptide sequences corresponding to METTL2A, METTL2B and METTL6 were detected in the respective Strep-METTL2A, 2B and 6 purifications (Fig. 1b). In addition, both the METTL2A and 2B purifications contained peptides corresponding to an uncharacterized protein encoded by the *DALR anticodon-binding protein 3* (*DALRD3*) gene (Fig. 1b, Supplementary Data 1). The molecular weights of the two predicted DALRD3 isoforms are 55 and 59 kDa, which correspond in size to the unidentified bands within the doublet observed by silver stain. For the METTL6 purification, we detected numerous peptides for seryl-tRNA synthetase (SARS) (Fig. 1b, Supplementary Data 1). The molecular weight of SARS (59 kDa) matches closely in size to the 60 kDa band detected by silver stain in the METTL6 purification. Moreover, SARS was recently identified as an interactor of METTL6[32]. However, no peptides corresponding to DARLD3 were identified in the METTL6 purification. These results suggest that DALRD3 forms a unique interaction with METTL2A and 2B.

To confirm the DALRD3 interaction and its specificity, we transiently expressed and purified METTL2A, METTL2B, and METTL6 followed by probing with antibodies against DALRD3. Immunoblot analysis revealed the co-purification of endogenous DALRD3 with Strep-METTL2A or METTL2B that was not readily apparent with Strep-METTL6 (Fig. 1c). As additional evidence that METTL2A and B interact with DALRD3, we also

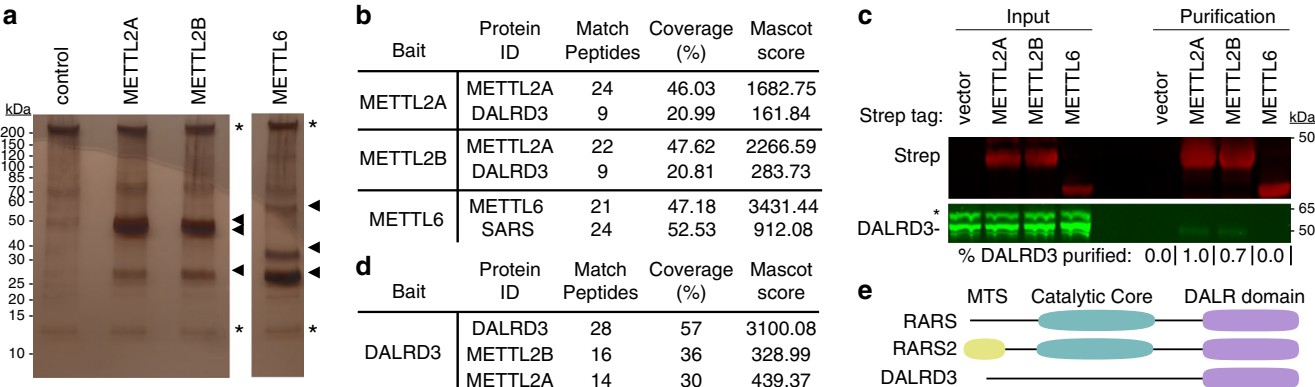

**Fig. 1 Human METTL2A/B interacts with DALR anticodon-binding domain containing 3 (DALRD3). a** Silver stain analysis of purified Strep-METTL2A, 2B and 6 from human 293T cells. Arrowheads denote bands that are enriched in the METTL2A, 2B and 6 purifications relative to control. Asterisks denote non-specific contaminants found in all three purifications. **b** Protein matches detected through LC-MS analysis of peptides present in the indicated protein purifications. **c** Immunoblot analysis of control, METTL2A, METTL2B and METTL6 purifications. The immunoblot was probed with anti-TwinStrep and anti-DALRD3 antibodies. Input represents 1% of total. Asterisk represents non-specific band. Percentage DALRD3 purified represents the fraction of DALRD3 recovered from the total input in each purification. N = 1. **d** Coverage of the reciprocal purification of DALRD3 in protein samples analyzed by LC-MS. **e** Schematic of human cytosolic arginyl-tRNA synthetase (RARS), mitochondrial arginyl-tRNA synthetase (RARS2) and DALRD3. Domains are indicated representing mitochondrial targeting signal (MTS), aminoacyl synthetase catalytic core and DALR anticodon-binding domain. Source data are provided as a Source data file.

performed a reciprocal purification of DALRD3 from human 293T cells stably expressing Strep-tagged DALRD3. Using LC-MS analysis, we detected multiple unique peptide matches to METTL2A and 2B in the DALRD3 purification indicative of an association between DALRD3 and endogenous METTL2A/B (Fig. 1d, Supplementary Data 2). These results identify DALRD3 as an interacting partner of METTL2A and 2B.

DALRD3 is an uncharacterized protein containing a carboxy-terminal sequence homologous to the "DALR" anticodon-binding domain found in arginyl tRNA synthetases from Archaea to Eukaryotes[36]. The DALR domain is named after characteristic conserved amino acids present in the primary protein sequence and folds into an all alpha helical structure as observed in *S. cerevisiae* arginyl tRNA synthetase[36–38]. In contrast to the carboxy-terminus, the amino-terminal portion of DALRD3 contains no recognizable motifs or domains and is specific only to DALRD3 homologs. While no DALRD3 homologs have yet been identified in single-celled eukaryotes or invertebrates, DALRD3 homologs can be detected in all sequenced vertebrates ranging from primitive jawless fishes to mammals[39,40]. Within vertebrates, the canonical DALR domain is found in cytoplasmic arginyl-tRNA synthetase 1 (RARS1) and mitochondrial arginyl-tRNA synthetase 2 (RARS2) in addition to DALRD3 (Fig. 1e). Unlike RARS1 and RARS2, however, DALRD3 lacks a recognizable tRNA synthetase catalytic motif, thereby suggesting a distinct function for DALRD3 outside of tRNA aminoacylation.

**The METTL2-DALRD3 complex binds distinct arginine tRNAs.** Based upon the structure of *S. cerevisiae* arginyl-tRNA synthetase, the DALR domain forms a nucleic acid binding pocket that recognizes the minor groove side of arginine tRNA anticodon stems through van der Waals, hydrophobic and electrostatic interactions[37]. Thus, we investigated whether DALRD3 interacts with tRNAs and if so, whether there was a particular binding specificity for DALRD3. To elucidate the RNA interactions mediated by DALRD3, we transiently expressed and purified Strep-DALRD3 from 293T cells either alone or with FLAG-METTL2A/B (Fig. 2a). Following binding to streptactin resin, we confirmed the purification of Strep-DALRD3 along with the co-purification of METTL2A or 2B (Fig. 2a). These results further

corroborate the interaction between DALRD3 and METTL2A/B as shown above.

We next examined the RNA species that co-purified with DALRD3 by denaturing PAGE followed by nucleic acid staining. While no RNAs were enriched in the control purifications, the Strep-DALRD3 purification contained several co-purifying RNA species that correspond in size to 5S and 5.8S rRNA along with tRNAs (Fig. 2b, lane 10). Of note, we observed a considerable reduction in co-purification of the 5S and 5.8S rRNAs with DALRD3 when co-expressed with either METTL2A or 2B (Fig. 2b, compare lane 10 to lanes 11 and 12). However, the co-purification of tRNAs with DALRD3 was maintained upon co-expression with METTL2A or 2B. These results indicate that DALRD3 interacts with rRNA and tRNAs when over-expressed and purified alone but shifts to predominantly binding tRNAs when assembled into a complex with METTL2A/B.

To determine the tRNA-binding specificity of DALRD3, we probed the DALRD3 purifications for distinct tRNAs via Northern blotting. Notably, we found that DALRD3 purifications were greatly enriched for arginine tRNAs known to contain m3C generated by METTL2A/B (Fig. 2c, lanes 10–12, tRNA-Arg-CCU and UCU). In contrast, arginine tRNA isoacceptors lacking m3C exhibited considerably less co-purification with DALRD3 (Fig. 2c, lanes 10–12, tRNA-Arg-UCG, CCG, and ACG). Consistent with the predicted specificity of the DALR domain for arginine tRNAs, only background levels of tRNA-Thr-AGU or Ser-UGA was present in any of the DALRD3 purifications whether co-expressed with or without METTL2A/B (Fig. 2c, Thr-AGU and Ser-UGA, lanes 10–12). Collectively, these results uncover a distinct tRNA-binding specificity for DALRD3 that contrasts with arginyl-tRNA synthetases, which recognizes all arginine tRNA isoacceptors regardless of the anticodon.

**DALRD3 facilitates METTL2A/B binding to tRNA-Arg-CCU and UCU.** The co-purification of DALRD3 with METTL2A and 2B along with the distinct tRNA-binding specificity of DALRD3 suggests that DALRD3 plays a role in targeting MET-TL2A/B to specific arginine tRNAs for m3C modification. To test this hypothesis, we transiently expressed Strep-METTL2A or 2B with or without co-expression of FLAG-tagged DALRD3 in 293T cells. Immunoblot analysis of the input extracts confirmed

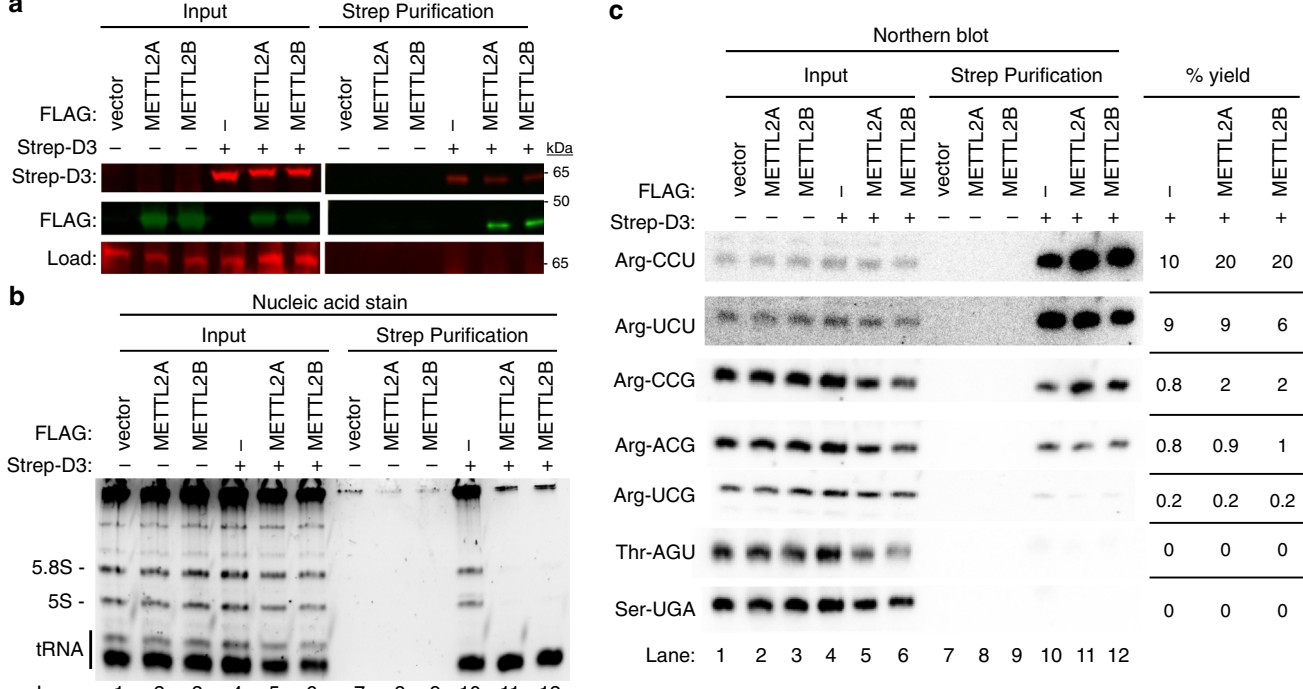

**Fig. 2 DALRD3 forms a complex with METTL2A/B to bind distinct arginine tRNAs. a** Immunoblot of Strep-DALRD3 purified from 293T cells expressed alone or in conjunction with FLAG-METTL2A/B. Immunoblot was probed with anti-TwinStrep and anti-FLAG antibodies. Load represents a non-specific protein band detected by the anti-Strep antibody used as load control. **b** Nucleic acid stain of RNAs extracted from the indicated input or purified samples after denaturing PAGE. The migration pattern of 5.8S rRNA (~150 nt), 5S rRNA (~120 nt) and tRNAs (~70–80 nt) are denoted. **c** Northern blot analysis of the gel in **b** using the indicated probes. Input represents 2% of total extracts used for purification. The percentage yield represents the amount of RNA in the Strep purification that was recovered from the total input. The experiment was performed three times with comparable results. Quantification was performed on the blot shown. Source data are provided as a Source data file.

the expression of Strep-METTL2A/B either alone or with FLAG-DALRD3 (Fig. 3a, lanes 2, 3, 5, and 6). Due to the transfection procedure, METTL2A and 2B exhibited reduced levels of expression when co-expressed with DALRD3 compared to METTL2A or 2B expressed alone. Using this approach, we detected the co-purification of FLAG-DALRD3 with Strep-METTL2A or 2B, further corroborating our finding that MET-TL2A/B interacts with DALRD3 (Fig. 3a, lanes 11 and 12). We also find that the interaction between DALRD3 and METTL2A/B persists after RNase treatment, suggesting that METTL2 interaction with DALRD3 is not bridged through RNA (Supplementary Fig. 2).

Analysis of copurifying RNAs revealed the co-purification of tRNA with both METTL2A and 2B (Fig. 3b, lanes 8–9 and 11–12). Moreover, there was an increase in the amount of tRNAs co-purifying with METTL2A or 2B when each was co-expressed with DALRD3 (Fig. 3b, compare lanes 8 and 9 with 11 and 12). The increase in co-purifying tRNAs with METTL2A/B when co-expressed with DALRD3 is not due to an increase in METTL2A/B expression or purification since there was actually less METTL2A/B expressed and purified in the presence of DALRD3 (Fig. 3a, lanes 11 and 12).

Using Northern blot hybridization, we detected low levels of tRNA-Arg-CCU and UCU copurifying with METTL2A or B alone but not tRNA-Arg-CCG, ACG or UCG (Fig. 3c, lanes 8 and 9). In addition, we found that tRNA-Thr-AGU also copurified with METTL2A or 2B when expressed alone (Fig. 3c, lanes 8 and 9, Thr-AGU). As a comparison, we detected only background levels of tRNA-Ser-UGA, tRNA-Gln-UUG or U6 snRNA in any of the purifications (Fig. 3c, lanes 7–12). The interaction of METTL2A and 2B with tRNA-Thr-AGU, Arg-CCU and Arg-

UCU but not the other tRNA-Arg isoacceptors nor tRNA-Ser-UGA is consistent with the substrate specificity of METTL2A/B[32]. Remarkably, the amount of tRNA-Arg-CCU or UCU was greatly increased in the METTL2A or 2B purification when co-expressed with DALRD3 (Fig. 3c, tRNA-Arg-CCU or UCU, compare lanes 8 and 9 with lanes 11 and 12). In contrast, the interaction between METTL2A or 2B with tRNA-Thr-AGU was disrupted by co-expression with DALRD3 (Fig. 3c, compare lanes 8 and 9 with lanes 11 and 12).

The enhancement of METTL2A/B interaction with arginine tRNAs by DALRD3 co-expression provides evidence that assembly of a METTL2-DALRD3 complex facilitates the recognition and targeting of specific arginine tRNA substrates for m3C modification. The slight co-purification of tRNA-Arg-CCU or UCU with METTL2A/B when expressed alone may be due to a minor population of METTL2A/B forming a complex with endogenous DALRD3, consistent with our original identification of DALRD3 with METTL2A/B via mass spectrometry. Moreover, the enrichment for tRNA-Arg-CCU and -UCU combined with the concomitant reduction in tRNA-Thr-AGU binding by METTL2A/B suggests that interaction with DALRD3 restricts the tRNA-binding specificity of METTL2A/B towards particular tRNA-Arg isoacceptors.

The tRNAs stably associated with METTL2 and/or DALRD3 could represent substrates that have not yet been modified with m3C and/or have been modified but not dissociated. To ascertain the relative m3C modification state of copurifying tRNAs, we used an oligonucleotide probe that displays differential hybridization based upon m3C modification status (described in further detail below). Using this probe, we found that DALRD3 co-purifies with both modified and unmodified tRNA-Arg-UCU and

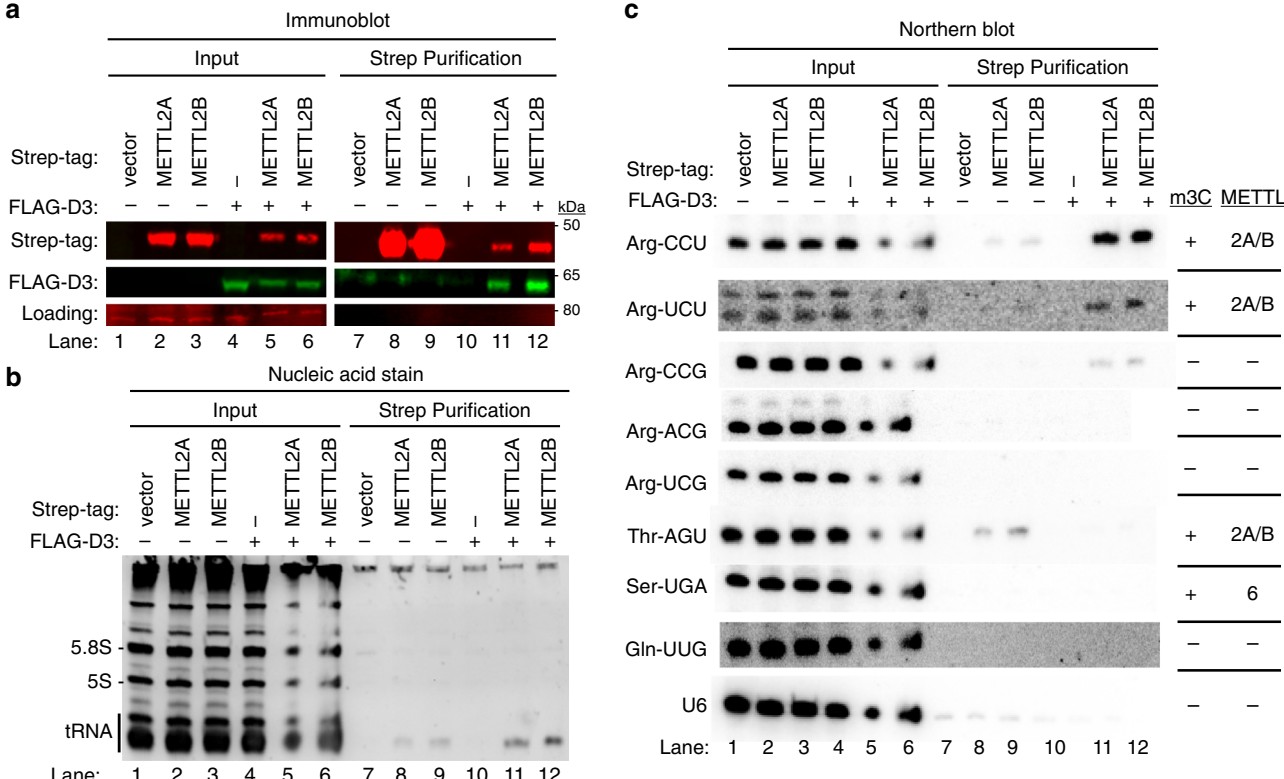

**Fig. 3 DALRD3 mediates the binding of METTL2A/B to tRNA-Arg-CCU and UCU. a** Immunoblot analysis of the purification of Strep-METTL2A/B expressed alone or with FLAG-DALRD3 from 293T human embryonic cells. The immunoblot was probed with anti-TwinStrep and anti-FLAG antibodies. Load represents a non-specific protein band detected by the anti-Strep antibody used as load control. Input represents 2% of total. **b** Nucleic acid stain of RNAs extracted from the indicated input or purified samples after denaturing PAGE. The migration pattern of tRNAs (70–80 nt), 5.8S rRNA (~150 nt), and 5 S rRNA (~120 nt) are denoted. **c** Northern blot analysis of METTL2A/B-associated RNAs with the indicated probes. The known presence or absence of m3C and the METTL enzyme that generates m3C in a given tRNA is denoted on the right. The purification was repeated three times with similar results. Source data are provided as a Source data file.

Arg-CCU (Supplementary Fig. 3a). In contrast, we found that the majority of tRNA-Arg-UCU and Arg-CCU that copurified with METTL2 was modified with m3C (Supplementary Fig. 3b). These results are consistent with DALRD3 protein binding unmodified tRNA-Arg-UCU or CCU for subsequent methylation upon interaction with METTL2.

**tRNA sequence requirements for methylation by METTL2-DALRD3.** Mammalian genomes express five different tRNA isoacceptors that decode arginine codons but only tRNA-Arg-CCU and Arg-UCU are modified to contain m3C at position 32 (Fig. 4a)[17,20,41]. Intriguingly, tRNA-Arg-UCG and Arg-ACG also contain C32 but are not modified by METTL2A/B in either human or mouse cells. Inspection of the isoacceptor stem loops reveals that tRNA-Arg-CCU and Arg-UCU contain U-A at positions 36 and 37 while tRNA-Arg-CCG, UCG, and ACG contain G-G at positions 36 and 37 (Fig. 4a). We also note that tRNA-Thr-AGU, which is a substrate of METTL2A/B, also contains a U-A dinucleotide at positions 36 and 37 (Fig. 4a, Thr-AGU). Moreover, A37 is modified to N6-threonylcarbamoyladenosine (t6A) in tRNA-Arg-CCU and Arg-UCU along with tRNA-Thr-AGU. Previous studies in *S. cerevisiae* and *S. pombe* have shown that the identity of residues in the anticodon loop along with the i6A/t6A modification at position 37 play key roles in the recognition and modification of seryl- and threonyl-tRNAs in m3C formation by the yeast Trm140 enzyme[17,25]. Combined with the tRNA interaction studies described above, these observations suggest

the possibility that DALRD3 recognizes specific arginine tRNAs, in part, through sequence elements in the anticodon loop that include positions 36 and 37 to facilitate METTL2A-dependent methylation.

To test this hypothesis, we probed the methyltransferase activity of METTL2-DALRD3 complexes purified from human cells on in vitro transcribed tRNA-Arg-CCU substrates. For monitoring m3C formation, we used a primer extension assay in which the presence of m3C leads to a reverse transcriptase (RT) block at position 32 while the lack of m3C allows for read-through and generation of an extended product. As a control to ensure that methylation was occurring at the correct position, we tested a tRNA-Arg-CCU substrate in which C32 was mutated to A (C32A). To probe the requirement for positions 36 and 37 in METTL2-DARLD3 methylation, we generated the tRNA-Arg-CCU substrates U36G and A37G (Fig. 4b).

As a positive control, we performed primer extension on RNA harvested from human 293T cells which results in an extension stop at position 32 of tRNA-Arg-CCU indicative of the m3C modification that was absent when no RT was added (Fig. 4c, lanes 1 and 2). No RT block at position 32 was detected for in vitro transcribed tRNA-Arg-CCU when pre-incubated with either water or negative control purification (Fig. 4c, lanes 3 and 4). In contrast, pre-incubation of tRNA-Arg-CCU with either purified METTL2A or 2B with or without DALRD3, followed by primer extension, revealed the appearance of a RT block at position 32 indicative of m3C formation (Fig. 4c, lanes 5–8). We

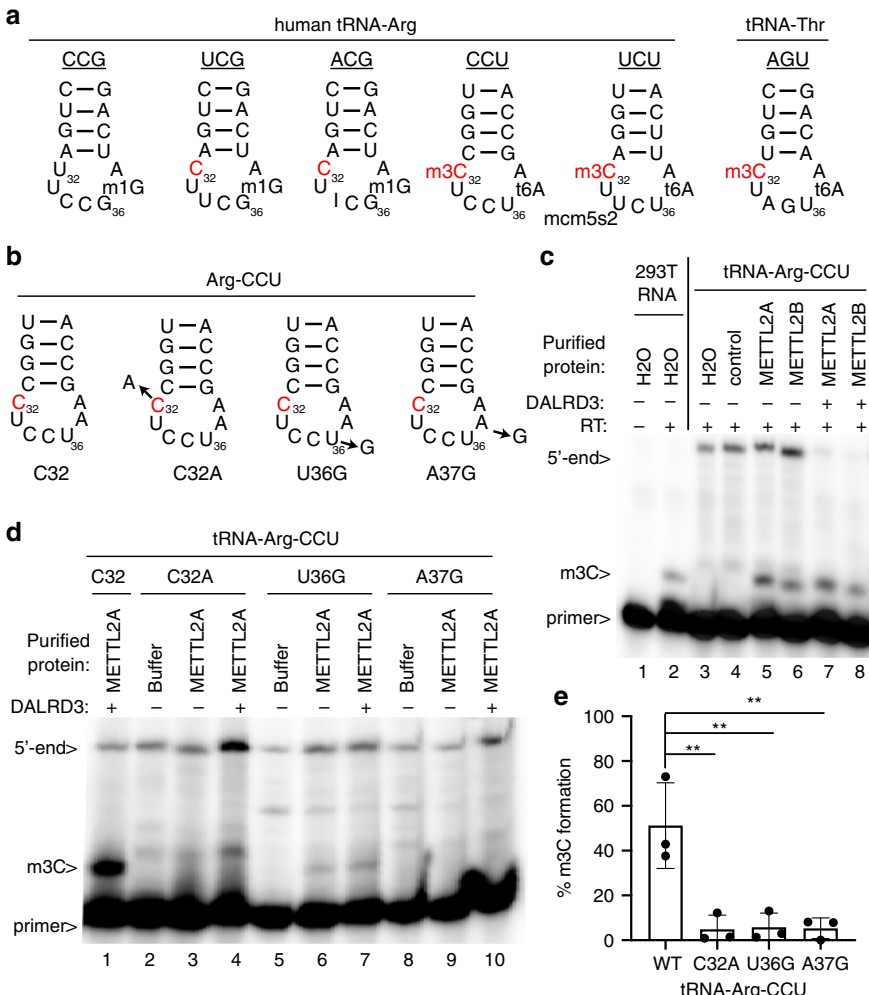

**Fig. 4 The identity of residues 36 and 37 in tRNA-Arg-CCU play a key role in m3C formation by METTL2A/2B in vitro. a** Anticodon loops of human tRNA-Arginine isoacceptors and known modifications. **b** Anticodon loops of in vitro transcribed tRNA-Arg-CCU and variants used in **c**, **d**. **c**, **d** Primer extension analysis of tRNA-Arg-CCU and variants incubated with either water, vector control eluate, purified METTL2A or METTL2B co-expressed with DALRD3. Primer length, 23 nt. **e** Quantification of m3C formation in **d**. Primer extension analysis was performed greater than three times for **c** and repeated three times for **d**. Bars represent the standard deviation from the mean. Statistical analysis was performed using one-way ANOVA and significance calculated using Tukey's multiple comparison test. **$P < 0.01$. $P = 0.0034$ for WT versus C32A, 0.0039 for WT versus U36G, 0.0037 for WT versus A37G. Source data are provided as a Source data file.

note that METTL2A/B purified from human cells without overexpression of DALRD3 exhibited m3C modification activity, consistent with the co-purification of endogenous DALRD3 with the over-expressed METTL2A or 2B to form an active methyltransferase complex. Thus, the purified METTL2-DALRD3 complex is active for m3C formation on an in vitro transcribed tRNA substrate lacking any other modifications.

Using the tRNA-Arg-CCU variants, we detected no RT block when C32 was mutated to A, providing evidence that position 32 was the site of methylation (Fig. 4d, lanes 1–4). Notably, mutation of either U36 or A37 to a G residue in tRNA-Arg-CCU led to a major decrease in m3C formation (Fig. 4d, lanes 5–10, quantified in 4e). These studies provide evidence that DALRD3 serves as a discrimination factor to recognize distinct arginine tRNAs based upon sequence elements common between tRNA-Arg-CCU and tRNA-Arg-UCU.

**m3C formation in cellular arginine tRNAs requires DALRD3.** The above results suggest that DALRD3 functions in the

recognition and interaction of specific arginine tRNAs with METTL2A/B. To investigate the requirement for DALRD3 in m3C formation in vivo, we engineered human cell lines lacking DALRD3 using CRISPR/Cas9 gene editing. Using the HAP1 human haploid cell line[42], we generated a cell clone containing a 14 base-pair deletion in exon 1 of the *DALRD3* gene (Supplementary Fig. 4a). The deletion is predicted to produce a truncated DALRD3 missing the majority of the polypeptide or a translation frameshift that results in nonsense mediated decay (NMD). Indeed, immunoblotting revealed the absence of full-length DALRD3 protein in the DALRD3-knockout (D3-KO) cell line compared to the isogenic control wildtype (WT) cell line (Fig. 5a). While expression of DALRD3 was abolished in the D3-KO cell line, no change in METTL2A levels was detected between the WT or D3-KO cell lines (Supplementary Fig. 4b).

To monitor m3C formation in tRNA, we used the positive hybridization in the absence of modification (PHA) assay (Fig. 5b)[43–46]. This Northern blot-based assay relies on differential probe hybridization to tRNA caused by the presence or absence of m3C, which impairs base-pairing[47–49]. Thus, a

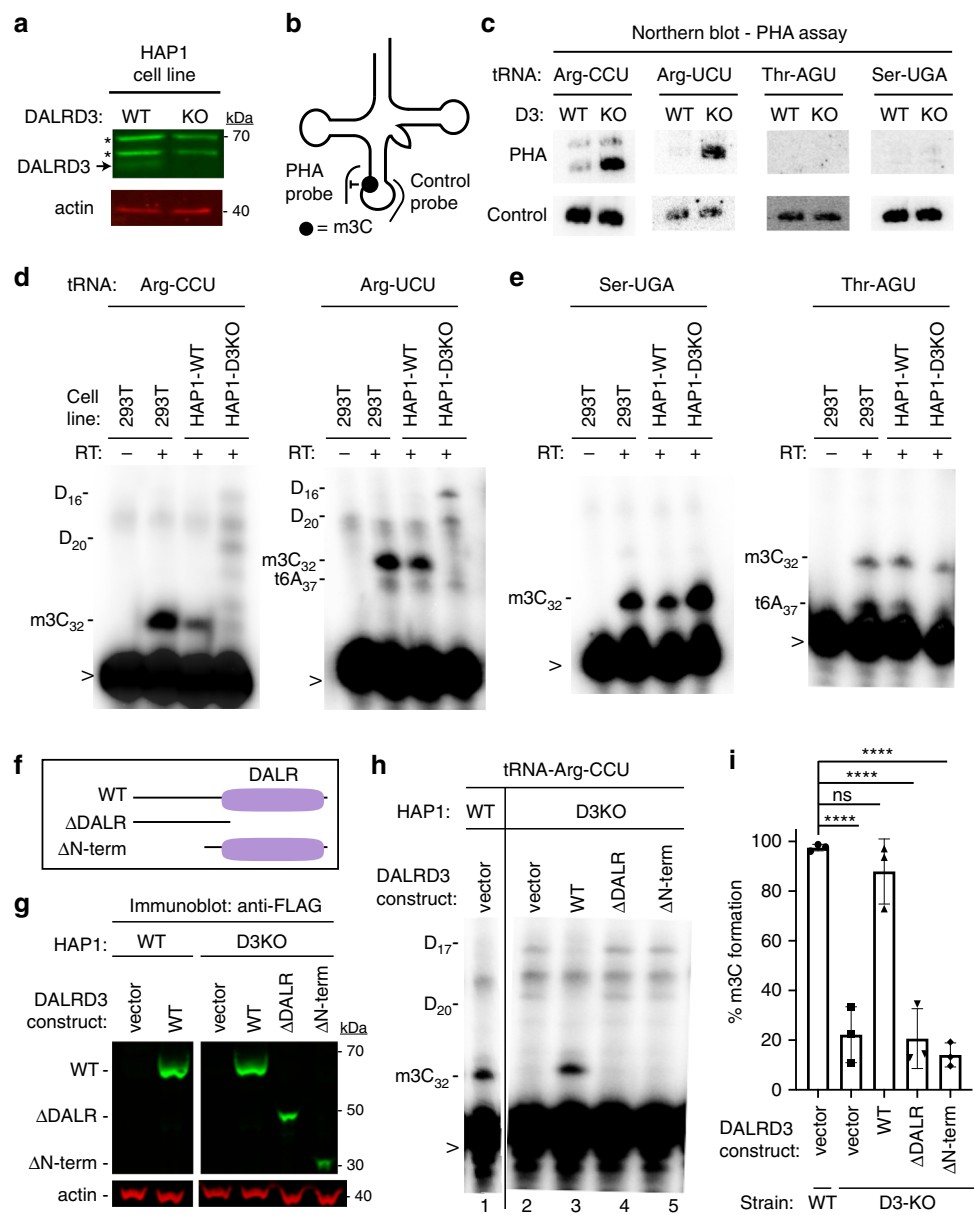

**Fig. 5 DALRD3 is required for efficient m3C formation in arginine tRNAs in vivo. a** Immunoblot verification for the loss of DALRD3 expression in human DALRD3-knockout (KO) cell lines compared to wild-type human HAP1 cells. Actin was used as a loading control. Asterisk denotes a non-specific band found in both wild-type and D3-KO cell lysates. **b** Schematic of the Positive Hybridization in the Absence of Modification (PHA) assay. **c** Northern blot analysis of PHA probes designed to detect m3C at position 32 and a control probe that hybridizes to a different area of the same tRNA. **d** Primer extension analysis of the tRNA-Arg-UCU and Arg-CCU harvested from the denoted human HAP1 cell lines. Length of primers; tRNA-Arg-UCU, 24 nt; tRNA-Arg-CCU, 23 nt. **e** Primer extension analysis of the tRNA-Ser-UGA and Thr-AGU harvested from the denoted human HAP1 cell lines. Length of primers; tRNA-Ser-UGA, 23 nt; tRNA-Thr-AGU, 20 nt. **f** Schematic of DALRD3 variants used for DALRD3 rescue experiments. **g** Immunoblot analysis confirming expression of DALRD3 variants in the indicated HAP1 cell lines. **h** Primer extension analysis of tRNA-Arg-CCU from WT or D3KO HAP1 cell lines stable integrated with the indicated DALRD3 expression constructs. (-RT) represents no reverse transcriptase was added; m3C- 3-Methylcytidine; D-dihydrouridine; t6A- threonylcarbamoyladenosine; > labelled probe. **i** Quantification of m3C formation in tRNA-Arg-CCU by primer extension. $n = 3$. Error bars represent standard deviation from the mean. Statistical analysis was performed using one-way ANOVA and significance calculated using Tukey's multiple comparison test. ****$P < 0.0001$; ns, non-significant. $P = 0.7430$ for WT strain + vector versus D3-KO strain + WT-DALRD3. **a**, (**c** through **e**), **g**, **h** were repeated three times each with similar results. Source data are provided as a Source data file.

decrease in m3C modification leads to an increase in PHA probe signal that can be normalized against the probe signal from a different region of the same tRNA as an internal control. For tRNA-Arg-CCU and UCU, we observed a considerable increase in PHA probe signal in the human D3-KO cell line compared to WT, indicating the loss of m3C modification in these particular tRNAs (Fig. 5c). In contrast, no increase in PHA signal was

detected for either tRNA-Ser-UGA or tRNA-Thr-AGU in the D3-KO cell line, consistent with the substrate specificity of DALRD3 shown above. We also note that the steady-state levels of all tested tRNAs remained similar between the WT and D3-KO cell lines, including tRNA-Arg-CCU and UCU.

As additional validation, we monitored m3C formation in specific cellular tRNAs using the primer extension assay. As

expected, an RT stop indicative of m3C at position 32 of tRNA-Arg-CCU and Arg-UCU was detected in wildtype human 293T and HAP1 cells (Fig. 5d). In contrast, the m3C block in both tRNA-Arg-CCU and Arg-UCU was greatly reduced in the D3-KO cell line with subsequent readthrough to the next RT-blocking modification (Fig. 5d, HAP1-D3KO lanes). Consistent with a role for DALRD3 only in arginine tRNA modification, the m3C modification in tRNA-Ser-UGA and Thr-AGU was unaffected in the D3-KO cell line (Fig. 5e). These results demonstrate that DALRD3 is required for efficient m3C formation in specific arginine tRNAs.

To dissect the regions of DALRD3 that play a role in m3C formation, we tested the ability of DALRD3 variants to rescue m3C formation in the HAP1-D3KO cell line. We generated stable cell lines expressing either wildtype DALRD3 or DALRD3 variants lacking either the DALR domain or the N-terminal extension (Fig. 5f, WT, ΔDALR, and ΔNterm). The expression of each DALRD3 variant in the stable cell lines was confirmed by immunoblotting (Fig. 5g). As expected, HAP1-D3KO cells with vector alone exhibited no m3C stop at position 32 in tRNA-Arg-CCU when compared to HAP1-WT cells (Fig. 5h, compare lanes 1 and 2). Re-expression of wildtype DALRD3 in HAP1-D3KO cells was able to rescue m3C formation in tRNA-Arg-CCU as evidenced by the re-introduction of the m3C RT block and the absence of read-through products (Fig. 5h, lane 3). In contrast, expression of the DALRD3 mutant lacking the DALR domain or the N-terminal extension was unable to restore m3C formation in tRNA-Arg-CCU (Fig. 5h, lanes 4 and 5; quantified in 5i). Altogether, these results demonstrate that m3C formation at position 32 in specific arginine tRNAs requires the DALRD3 protein with both the DALR domain and N-terminal region contributing key roles in methyltransferase activity.

**Impact of m3C modification on codon reporter expression.** The interaction between nucleotides 32 and 38 of the tRNA anticodon loop is known to impact codon recognition and tRNA binding to the ribosome in *Escherichia coli*[15]. Thus, we investigated the possible effects of m3C$_{32}$ modification on tRNA-Arg function in protein synthesis using codon-dependent reporter constructs. We generated lentiviral reporters expressing nanolu-ciferase protein fused downstream of ten consecutive AGA or AGG codons, which are decoded by m3C-containing tRNA-Arg-UCU or CCU (Supplementary Fig. 5a). We also generated reporter plasmids containing codon runs of CGA or CGC, which are decoded by tRNA-Arg-UCG or ACG lacking m3C. Based upon this system, we found no significant difference in protein expression between the WT or D3-KO cell lines for any of the arginine codon reporters (Supplementary Fig. 5b and c). We also detected no major change in expression of a control GFP construct between WT and D3-KO cell lines (Supplementary Fig. 5b and c). These results suggest that the m3C modification has minimal effect on translation efficiency in these codon reporters and/or plays a role in arginine tRNAs that cannot be detected using this experimental system.

**A *DALRD3* variant linked to a developmental brain disorder.** Due to the emerging links between tRNA modification and neurodevelopmental processes, we next investigated whether DALRD3 is associated with any neurological disorders of unknown genetic etiology. We identified a consanguineous family with two sibling patients exhibiting profound global developmental delay and persistent early infantile epilepsy who tested negative for known disease mutations by routine karyotype, chromosomal microarray, and an epilepsy gene panel (Fig. 6a). The family was identified by applying a "genomics first" approach

to uncharacterized patients with neurodevelopmental disorders[50]. Parents are healthy first cousins and they have one healthy child in addition to the two affected children and one spontaneous miscarriage (Fig. 6b). Subsequent exome sequencing guided by autozygome analysis[51] revealed a single homozygous nucleotide substitution resulting in a C to A transversion in exon 9 of the *DALRD3* gene in a shared region of autozygosity between the two affected siblings on chromosome 3 (Fig. 6c). Confirmatory Sanger sequencing validated the full segregation of this variant with the disease in the family in a fully penetrant autosomal-recessive model, being homozygous in both patients and heterozygous in the parents and the healthy sibling (Fig. 6d). The genetic altera-tion is expected to cause a nonsense mutation due to the intro-duction of a UAA stop codon within the mRNA transcript encoding the predominant isoform of DALRD3 (pTyr417*). Translation of this transcript will result in a truncated protein lacking the DALR tRNA anticodon-binding domain. In addition, the variant *DALRD3* mRNA could be subject to NMD due to the presence of a premature stop codon.

To determine the molecular impact of the DALRD3 mutation on m3C formation, we generated lymphoblastoid cell lines (LCLs) from the two sibling patients (referred to as D3-LCLs generated from patients 1 and 2; P1 and P2). LCLs generated from the affected patients were compared to control lymphoblasts from two ethnically matched, healthy, unrelated individuals (LCL-WT1 and WT2). Consistent with the nonsense mutation leading to a truncation of the DALRD3 protein and/or NMD, the levels of full-length DALRD3 protein were decreased in cell lysates prepared from either patient LCLs compared to WT-LCLs (Fig. 7a). Using the aforementioned PHA assay, we detected a substantial increase in PHA signal for tRNA-Arg-CCU and UCU in D3-LCLs compared to WT control cells, indicating the loss of m3C modification in these particular tRNAs (Fig. 7b). Follow up investigation using the primer extension assay confirmed the severe reduction of m3C modification in tRNA-Arg-CCU and UCU of both D3-LCLs from affected patients (Fig. 7c, d). In contrast, the levels of m3C modification detected by PHA probe hybridization or primer extension for tRNA-Ser-UGA and tRNA-Thr-AGU was similar between D3- and WT control-LCLs (Fig. 7b–d). The deficit in m3C modification in the arginine tRNAs of both patient LCLs demonstrates that the *DALRD3* mutation represents a partial loss-of-function allele.

Epileptic encephalopathies (EEs) encompass a large, hetero-genous group of early-onset neurodevelopmental disorders characterized by intractable seizures and electroencephalogram abnormalities (reviewed in refs. [52,53]). In both patients harboring the DALRD3 mutation, the onset of seizures occurred early in infancy at 6 to 7 months of age indicative of early-onset epileptic encephalopathy (Table 1). While the epilepsy of patient 2 could be controlled by anti-epileptic medications, the seizures in patient 1 were poorly controlled by medication. In addition to develop-mental delay and epilepsy, both patients exhibited a range of neurological and physiological symptoms consisting of severe motor and speech phenotypes, hypotonia and facial dysmorphia (Table 1). Based upon the constellation of symptoms characterized by the co-occurrence of developmental delay and frequent epileptic activity, both patients match clinical conditions now classified as developmental and epileptic encephalopathy[54]. Altogether, these findings identify human individuals who lack m3C modification in arginine tRNAs and reveal a biological link between m3C tRNA modification and proper neurological function.

## Discussion
Here, we show that human METTL2A/B forms a complex with the DALRD3 protein to catalyze the formation of m3C in specific

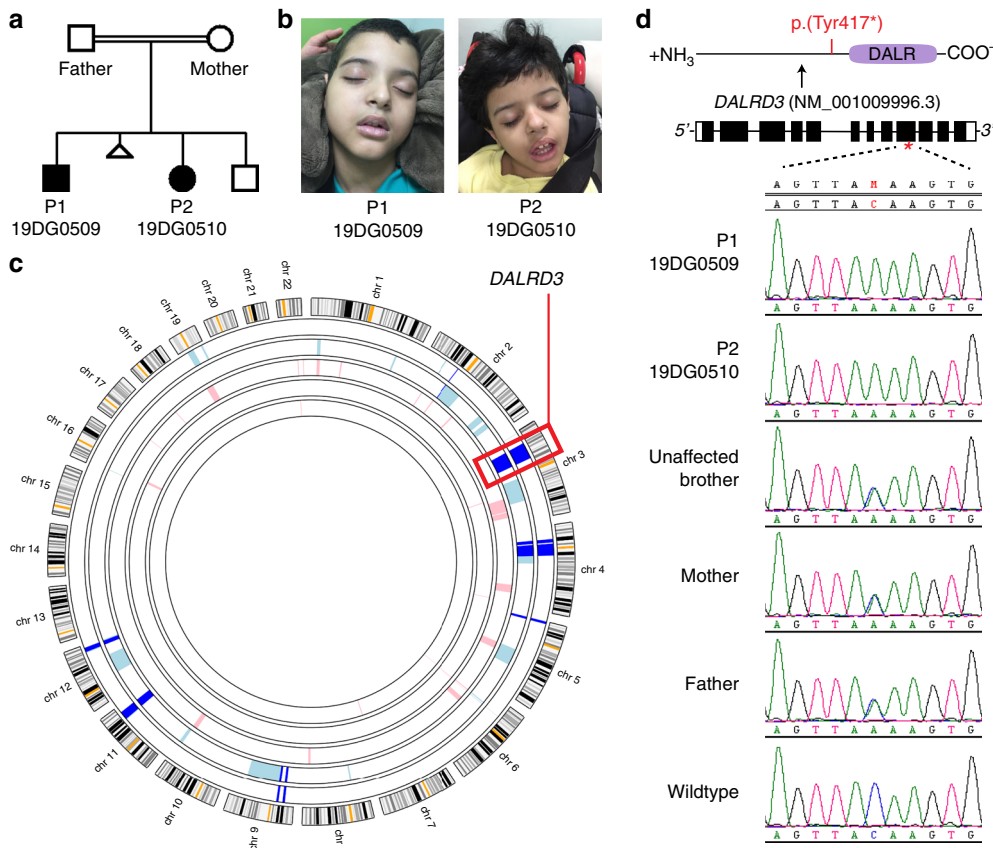

**Fig. 6 Identification of a *DALRD3* variant linked to developmental delay and epileptic encephalopathy. a** Pedigree of the family harboring a nonsense mutation in the *DALRD3* gene. **b** Patients 1 and 2 (P1 and P2) containing the homozygous *DALRD3* mutation. **c** Agile MultiIdeogram output of autozygosity analysis of the study family as a series of block arcs with the two affected individuals represented by the outer two arcs. Autozygous regions from affected individuals are marked as pale blue, those from unaffected individuals as pink, and those homozygous in all affected individuals as dark blue. **d** Sanger sequencing chromatograms of the indicated individuals from the family in this study along with a wildtype, control individual.

arginine tRNAs. Our findings explain how METTL2A/B recognizes particular arginine tRNAs for m3C modification through the unique tRNA interaction specificity of DALRD3 protein. Notably, we have identified human individuals with autosomal-recessive nonsense mutations in the *DALRD3* gene who lack m3C in their arginine tRNAs. These individuals exhibit a disorder characterized by severe developmental delay and early-onset epileptic encephalopathy, thereby linking the m3C modification in mammalian arginine tRNAs with proper neurological function.

The METTL2-DALRD3 complex presents yet another case of a multi-subunit enzyme catalyzing tRNA modification. The emerging number of multimeric tRNA modification enzymes has been hypothesized to be driven by the need to recognize and modify different substrates while maintaining high specificity[55,56]. In one pertinent example, the activity of *S. cerevisiae* Trm140p on seryl-tRNA substrates is greatly stimulated by interaction with seryl-tRNA synthetase, which can recognize the full cellular repertoire of seryl-tRNAs even though their anticodons do not have any nucleotides in common[25]. Human METTL6 also interacts with seryl-tRNA synthetase[32], providing further evidence that Trm140p homologs have evolved to bind additional protein cofactors in order to efficiently recognize their substrates. The interaction between human METTL2A/B and DALRD3 is analogous to the interaction of *S. cerevisiae* Trm140p with seryl-tRNA synthetase in that a tRNA methyltransferase interacts with a known or predicted tRNA-binding protein in order to catalyze tRNA modification. However, unlike the interaction of seryl-tRNA synthetase with Trm140p, which allows Trm140p to

recognize and modify different seryl-tRNA isoacceptors in *S. cerevisiae*, the tRNA-binding specificity of DALRD3 limits METTL2A/B activity to only tRNA-Arg-UCU and CCU. The need to discriminate between the different arginine tRNA isoacceptors could explain why mammals evolved DALRD3 to interact with METTL2 rather than arginyl-tRNA synthetase. Another possibility is that METTL2 contributes primarily to substrate specificity while DALRD3 serves as an activator of METTL2 methyltransferase activity.

An outstanding question that remains is why the m3C modification is present only in a subset of arginine tRNAs. One possibility is that tRNA-Arg-UCU and Arg-CCU possess additional requirements for folding and/or stability that are distinct from the remaining tRNA-Arg isoacceptors due to differences in anticodon structure. In addition, the m3C modification in tRNA-Arg-UCU and CCU could be required for their proper aminoacylation and/or interaction with the ribosome during translation. Interestingly though, DALRD3-deficient human cells exhibited no major change in the expression of reporter proteins fused to a run of AGA or AGG codons, both of which are decoded by arginine tRNAs containing m3C. However, the arginine codon reporter system represents a highly specific context that might not reflect the actual physiological impact of the m3C modification on tRNA function in translation in different tissues. Future studies using ribosome profiling approaches in different DALRD3-deficient cell types could reveal the mRNA transcripts that are dependent upon m3C modification for proper translation and protein expression.

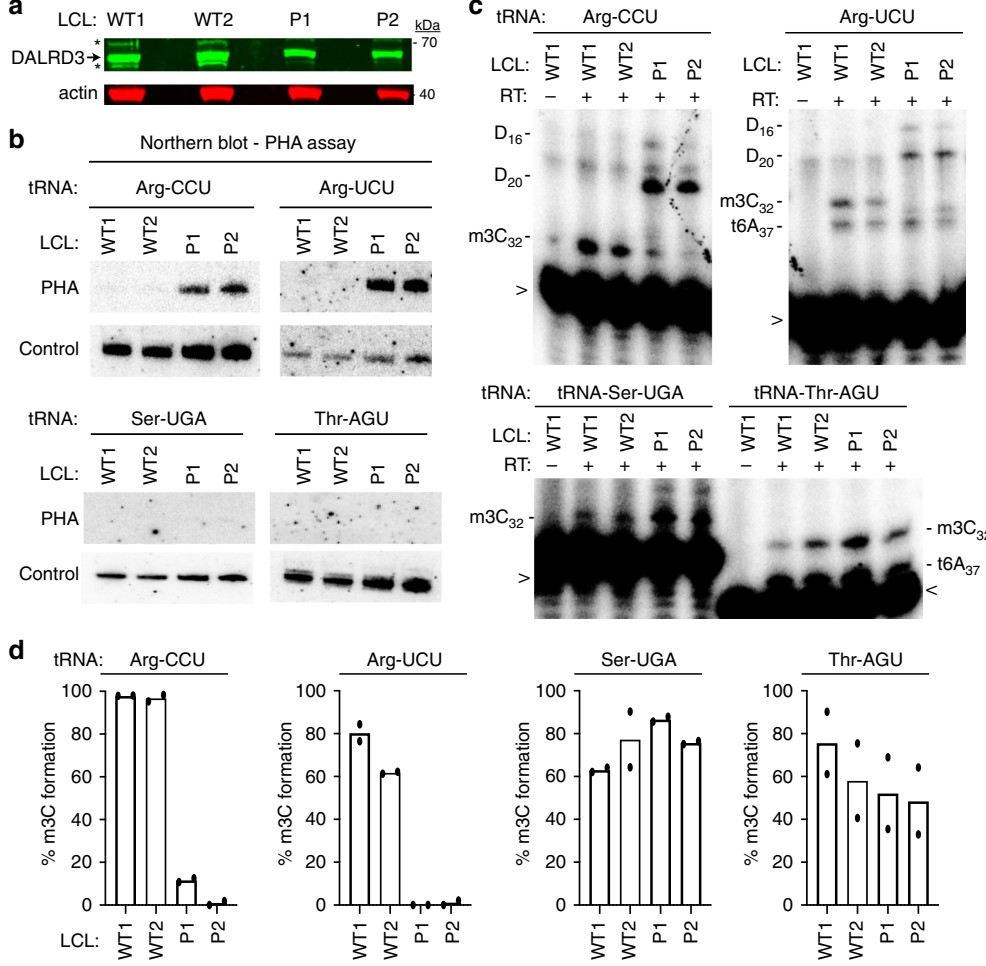

**Fig. 7 Lymphoblastoid cell lines derived from patients with the homozygous DALRD3 variant exhibit deficiency in m3C modification in arginine tRNAs.** **a** Immunoblot for the indicated proteins from lysates prepared from LCLs donated from wildtype (WT) individuals and patients P1 and P2 harboring the homozygous DALRD3 mutation. Asterisks (*) represent non-specific bands on either side of the DARLD3 protein. Blot was repeated twice with similar results. **b** Northern blot analysis using PHA probes designed to detect m3C at position 32 and a control probe that hybridizes to a different area of the same tRNA. Blot was repeated three times with comparable results. **c** Primer extension analysis of the indicated tRNAs from the denoted human LCLs. (-RT) represents no reverse transcriptase; m3C 3-Methylcytidine; D dihydrouridine; t6A threonylcarbamoyladenosine. > labelled probe. Primer extension analysis of lymphoblastoid cell lines was repeated three times. **d** Quantification of m3C formation by primer extension for the indicated tRNAs. $n = 2$. Source data are provided as a Source data file.

Notably, the appearance of DALRD3 in the vertebrate lineage coincides with the emergence of METTL2 homologs as well as the m3C modification in vertebrate arginine tRNAs. An intriguing possibility is that DALRD3 arose in vertebrates through a duplication event from an extant arginyl-tRNA synthetase gene that lost catalytic activity but retained tRNA binding through the DALR domain. DALRD3 could also have resulted from the insertion of the DALR domain coding sequence downstream of a pre-existing protein-coding gene to generate a vertebrate-specific fusion protein with a unique domain architecture. In either case, the utilization of the DALR anticodon-binding domain for tRNA modification highlights the modular nature of motifs linked to aminoacyl synthetases and the expansion of functions from homologous synthetase predecessors[57,58]. Future studies will determine if DALRD3 serves additional roles besides tRNA modification in vertebrates. For example, DALRD3 may interact with additional RNAs besides tRNAs to modulate their modification status or stability. We also note that m3C at position 47:3 of the variable arm is present in mammalian tRNA-Ser isoacceptors but not *S. cerevisiae* tRNAs[17], suggesting the possibility of another mammalian-specific tRNA methyltransferase complex that remains to be identified.

The identification of a *DALRD3* mutation that abolishes m3C formation in human patients with developmental delay and epileptic encephalopathy has uncovered a physiological link between the m3C modification and neurological function. This finding is intriguing because the majority of tRNA modification defects that are known to cause epileptic encephalopathies are due to alterations in mitochondrial function. For example, loss of the i6A[59,60] or wobble taurine modification[61–64] in mitochondrial tRNAs causes abnormal mitochondrial protein synthesis and defects in mitochondrial respiratory chain complexes that lead to developmental delay and epileptic encephalopathy. While there are no known mitochondrial tRNAs dependent upon DALRD3 for m3C formation, there could be a possible indirect impact of DALRD3-dependent tRNA modification on mitochondrial homeostasis since cytoplasmic tRNA modifications in the anticodon loop have been shown to impact mitochondrial metabolism[43]. The loss of arginine tRNA modification due to the DALRD3 mutation provides an avenue to explore additional biological pathways that are key to proper neurological function and perturbed in neurological disorders. Future experiments will investigate the role of DALRD3 and its associated tRNA modifications in protein

---

**Table 1 Clinical phenotype and molecular findings for patients with homozygous variants in DALRD3.**

| | Patient 1 | Patient 2 |
|---|---|---|
| ID | 19DG0509 | 19DG0510 |
| Gender | Male | Female |
| Date of birth | 17/04/2009 | 05/01/2013 |
| Genotype | NM_001276405.1:c.1251 C > A: | NM_001276405.1:c.1251 C > A: |
| Protein | p.(Tyr417*) | p.(Tyr417*) |
| Perinatal history | Normal spontaneous vaginal delivery with history of placental insufficiency and oligohydramnios | Full term product of cesarean section due to breech presentation and oligohydramnios in addition to placental insufficiency |
| Weight at birth (kg) | 2.25 (−2.2 SD) | 2.5 (4th centile) |
| Developmental delay | Severe | Severe |
| Motor | Immobile | Immobile |
| Speech | Nonverbal | Nonverbal |
| Seizures | Seizures started at age 7 months in the form of myoclonic jerks which remains frequent and poorly controlled by antiepileptic medications | At age 6 months epilepsy ensued, initially as brief episodes of flexion tonic spasm of head followed by myoclonic seizures. Unlike the sibling brother, the epilepsy of patient 2 is reasonably controlled by antiepileptic medications. |
| EEG | Independent multifocal epileptic discharges predominantly over the anterior head region bilaterally as well as over the right temporal and right parietal regions | Markedly high voltage and slow background for age along with slow generalized polyspike and wave activity |
| Tone | Axial and peripheral hypotonia with dystonic like movement and generalized muscle wasting | Central and peripheral hypotonia with dystonic like movements and generalized muscle wasting |
| Microcephaly | No | Yes |
| Brain finding | Mild diffuse brain parenchymal volume loss with diffuse paucity of the myelin within the brain parenchyma. | Normal topographical and morphological appearance of the infratentorial and supratentorial structures |
| Audiology assessment | Moderate to severe conductive hearing loss in the left ear and mild conductive hearing loss in right ear | N/A |
| Dysmorphism | Subtle facial dysmorphia and small left ear | Microcephaly with subtle facial dysmorphia |
| Other | Severe gastroesophageal reflux disease necessitating GT tube placement and fundoplication at age 4 years, no visual tracking or social smile | Vomiting and chocking on first day of life, mild congenital heart disease that resolved spontaneously, ectopic right kidney, bilateral optic disc pallor |

translation to identify cellular pathways that would be affected by the m3C modification and their roles in cell physiology and development. An increased understanding of the genetic and molecular underpinnings of early-onset epileptic encephalopathies will be especially vital since many types of these epilepsies are resistant to standard conventional epilepsy therapy[65].

## Methods

**Plasmids**. The open reading frame (ORF) for METTL2A and METTL2B was RT-PCR amplified from HeLa human cervical carcinoma cell cDNA, cloned by restriction digest and verified by Sanger sequencing. The cDNA clones for METTL2A and METTL2B correspond to NM_181725.3 and NM_018396.2, respectively. The ORF for DALRD3 was PCR amplified from cDNA plasmid HsCD00338640 (Plasmid Repository, Harvard Medical School) and cloned into either pcDNA3.1-TWIN-Strep or pcDNA3.1-3xFLAG. Truncation constructs were PCR amplified and cloned from the ORF of METTL2A and DALRD3 and cloned into pcDNA3.1-3xFLAG for METTL2A truncation constructs and DALRD3 truncation constructs. Primers used are listed in Supplementary Table 1.

For lentiviral expression constructs, Strep tagged-METTL2A, 2B and DALRD3 as well as 3x-FLAG-tagged DALRD3 and all truncation constructs of DALRD3 were first cloned into the pENTR.CMV.ON plasmid using NheI and NotI restriction sites. The entry vectors were recombined via LR clonase reaction (ThermoFisher) into the pLKO.DEST.puro destination vector to allow for stable integration of the target genes into human cell lines.

For the arginine codon reporter assay, a synthetic gene encoding the NanoLuc luciferase protein fused to an N-terminal run of 10X arginine codons of either AGA, AGG, CGC or CGA was generated by in vitro gene synthesis (GenScript). The synthetic codon reporter gene was subcloned into pcDNA3.1 and sequence verified. The reporter cassette containing the 10xArg codon-NanoLuc was then cloned into the pLJM1-EGFP lentiviral plasmid (Sabatini Lab, Addgene #19319) using NheI-EcoRI to replace the EGFP sequence. The resulting plasmid encodes the luciferase reporter protein expressed from the CMV promoter.

**Tissue cell culture and generation of stable cell lines**. The 293T human embryonic cell line was originally obtained from ATCC (CRL-3216). The HAP1 Human Male Chronic Myelogenous Leukemia (CML) cell line was obtained from Horizon Discovery Life Sciences. Human lymphoblastoid cell lines were generated

by EBV immortalization of primary human lymphocytes obtained from patient blood samples. 293T human embryonic kidney cell lines were cultured in Dulbecco's Minimal Essential Medium (DMEM) supplemented with 10% fetal bovine serum (FBS), 1X penicillin and streptomycin (ThermoFisher), and 1X Glutamax (Gibco) at 37 °C with 5% $CO_2$. Cells were passaged every 3 days with 0.25% Trypsin. Human HAP1 DALRD3 knock-out cell lines were generated by CRISPR/ Cas9 mutagenesis (Horizons Discovery Life Sciences). Human HAP1 cell lines were cultured in Iscove's Modified Dulbecco's Medium (IMDM) supplemented with 10% FBS and 1X penicillin and streptomycin at 37 °C with 5% $CO_2$. Cells were passaged with 0.05% Trypsin. Human lymphoblastoid cell lines were cultured in Roswell Park Memorial Institute (RPMI) 1640 Medium containing 15% fetal bovine serum, 2 mM L-alanyl-L-glutamine (GlutaMax, Gibco) and 1% Penicillin/ Streptomycin.

For generation of stable cell lines, $2.5 \times 10^5$ 293T cells were seeded onto $60 \times 15$ mm tissue culture dishes. In all, 1.25 µg of pLKO.DEST.puro plasmids containing the cloned ORF of METTL2A, 2B, 6, DALRD3, and DALRD3 truncations or empty vector along with a lentiviral packaging cocktail containing 0.75 µg of psPAX2 packaging plasmid and 0.5 µg of pMD2.G envelope plasmid was transfected into HEK293T cells using calcium phosphate transfection. In all, 48 hours after transfection, media containing virus was collected and filtered sterilized through 0.45 µm filters and flash frozen in 1 ml aliquots. Lentivirus for the pLJM-EGFP or codon reporter plasmids were generated in an identical manner.

For lentiviral infection in 293T or HAP1 cell lines, $2.5 \times 10^5$ cells were seeded in six-well plates or $6 \times 10^4$ cells in 24-well plates for HAP1 cells infected with the NLuc reporter plasmid. 24 hours after initial seeding, 1 ml of virus (or media for mock infection) along with 2 ml of media supplemented with 10 µg/ml of polybrene was added to each well. The cells were washed with PBS and fed fresh media 24 h post-infection. Puromycin selection begun 48 h after infection at a concentration of 2 µg/ml. Fresh media supplemented with puromycin was added every other day and continued until the mock infection had no observable living cells. Proper integration and expression of each construct was verified via immunoblotting.

**Transient transfections and protein-RNA purifications**. 293T cells were transfected via calcium phosphate transfection method[66]. Briefly, $2.5 \times 10^6$ cells were seeded on $100 \times 20$ mm tissue culture grade plates (Corning) followed by transfection with 10–20 µg of plasmid DNA. Cells were harvested 48 h later by trypsin and neutralization with media, followed by centrifugation of the cells at $700 \times g$ for 5 min followed by subsequent PBS wash and a second centrifugation step.

Protein was extracted by the Hypotonic Lysis protocol immediately after cells were harvested post-transfection. Cell pellets were resuspended in 0.5 μl of a hypotonic lysis buffer (20 mM HEPES pH 7.9, 2 mM MgCl₂, 0.2 mM EGTA, 10% glycerol, 0.1 mM PMSF, 1 mM DTT) per 100 × 20 mm tissue culture plate. Cells were kept on ice for 5 min and then underwent a freeze-thaw cycle three times to ensure proper detergent-independent cell lysis. NaCl was then added to the extracts at a concentration of 0.4 M and subsequently incubated on ice for 5 mins and spun down at 14,000 × g for 15 min at 4 °C. In all, 500 μl of Hypotonic Lysis buffer supplemented with 0.2% NP-40 was added to 500 μl of the supernatant extract.

TWIN-Strep tagged proteins were then purified by incubating whole cell lysates from the transiently-transfected cell lines with 50 μl of MagStrep "type3" XT beads (IBA Life Sciences) for two hours at 4 °C. Magnetic resin was washed three times in 20 mM HEPES pH 7.9, 2 mM MgCl₂, 0.2 mM EGTA, 10% glycerol, 0.1% NP-40, 0.2 M NaCl, 0.1 mM PMSF, and 1 mM DTT. Proteins were eluted with 1X Buffer BX (IBA LifeSciences) which contains 10 mM D-biotin. Purified proteins were visualized on a NuPAGE Bis-Tris polyacrylamide gel (ThermoFisher) and then transferred to Immobilon-FL Hydrophobic PVDF Transfer Membrane (Millipore Sigma) with subsequent immunoblotting against either the FLAG tag or TWIN-Strep tag (Anti-FLAG M2, Sigma-Aldrich; THE™ NWSHPQFEK antibody, GenScript). For immunoblots against endogenous DALRD3, Proteintech's DALRD3 antibody was used.

We also utilized this procedure followed by TRIzol RNA extraction directly on the beads in order to identify co-purifying RNA with each protein of interest. Beads first underwent three washes in the Lysis Buffer as mentioned previously and then resuspended in 250 μl of Molecular Biology Grade RNAse-free water (Corning). 10 μl of the bead-water mixture was taken for immunoblotting analysis where the beads were mixed with 2X Laemmeli Sample Buffer (Bio-Rad) supplemented with DTT and boiled at 95 °C for five minutes prior to loading onto a BOLT 4–12% Bis-Tris Plus gel (Life Technologies). RNA extraction followed TRIzol LS RNA extraction protocol (Invitrogen). RNA was resuspended in 5 μl of RNAse-free water and loaded onto a 10% polyacrylamide, 7 M urea gel. The gel was then stained with SYBR Gold nucleic acid stain (Invitrogen) to visualize RNA.

FLAG- purifications for the DALRD3 truncations followed the same protein purification protocol outlined above, however, we used Anti-DYKDDDDK Magnetic Beads (Clontech) as our purification resin. These purifications than underwent the TRIzol-RNA extraction procedure.

### Liquid chromatography-tandem mass spectrometry analysis.
Each stably-integrated cell line was plated on two 150 × 25 mm tissue culture dishes plates and grown until 70% confluency before cells of each line were combined and harvested for protein extraction via the aforementioned hypotonic lysis protocol. Proteins were again extracted on MagStrep "type 3" XT beads and washed in the same buffer as mentioned above. To ensure all protein was efficiently eluted off of the magnetic resin, the beads were left in 10 mM D-biotin (Buffer BX, IBA Life-Sciences) overnight at 4 °C. Two one-hour elutions were completed the following day and all elutions were pooled together for each individual sample. The total eluate was then placed on a Spin-X UF 500 μl Centrifugal Concentrator (Corning) and spun at 15,000 × g for approximately 1 hour and 15 min at 4 °C.

For protein separation, 30 μl of concentrated eluate was fractionated on a NuPAGE 4–12% Bis-Tris Protein gel (ThermoFisher). The gel was fixed overnight in 40% ethanol and 10% acetic acid. The gel was incubated in Sensitizing Solution (30% ethanol, 0.2% sodium thiosulphate and 6.8% sodium acetate) for 30 minutes before being washed three times with water for 5 min each wash. The gel was then stained in 0.25% silver nitrate for 20 min and washed twice more with water for 1 min each time. The bands were visualized by developing in 2.5% sodium carbonate and 0.015% formaldehyde and allowed to incubate until bands appeared. The remainder of each eluate (~65 μl) was loaded on a NuPAGE 4–12% Bis-Tris protein gel and briefly fractionated to yield a single gel band corresponding to the majority of proteins within each purification. Gel bands were excised using a razor blade and subject to in gel reduction, alkylation and trypsin digest by the URMC Mass Spectrometry Resource Lab[67,68].

Peptides were injected onto a homemade 30 cm C18 column with 1.8 μm beads (Sepax), with an Easy nLC-1000 HPLC (Thermo Fisher), connected to a Q Exactive Plus mass spectrometer (Thermo Fisher). Solvent A: 0.1% formic acid in water, Solvent B: 0.1% formic acid in acetonitrile. The gradient began at 3% B and held for 2 min, increased to 30% B over 13 min, increased to 70% over 2 minutes and held for 3 mins, then returned to 3% B in 2 min and re-equilibrated for 8 min, for a total run time of 30 min. For peptides isolated from the single gel band corresponding to the majority of proteins, the gradient began at 3% B and held for 2 min, increased to 30% B over 41 min, increased to 70% over 3 min and held for 4 min, then returned to 3% B in 2 min and re-equilibrated for 8 minutes, for a total run time of 60 minutes. The Q Exactive Plus was operated in data-dependent mode, with a full MS1 scan followed by 8 data-dependent MS2 scans. The full scan was done over a range of 400–1400 m/z, with a resolution of 70,000 at m/z of 200, an AGC target of 1e6, and a maximum injection time of 50 ms. The MS2 scans were performed at 17,500 resolution, with an AGC target of 1e5 and a maximum injection time of 250 ms. The isolation width was 1.5 m/z, with an offset of 0.3 m/z, and a normalized collision energy of 27 was used.

Raw data was searched using the SEQUEST search engine within the Proteome Discoverer software platform, version 1.4 (Thermo Fisher), using the SwissProt human database that was downloaded in December of 2015. Trypsin was selected as the enzyme allowing up to two missed cleavages, with an MS1 mass tolerance of 10 ppm, and an MS2 mass tolerance of 25 mmu. Carbamidomethyl was set as a fixed modification, while oxidation of methionine was set as a variable modification. Percolator was used as the FDR calculator, filtering out peptides which had a q-value > 0.01[69]. The Mascot scores of individual peptides were calculated as the absolute probability that the observed peptide match is a random event when matching spectra to all the expected spectra of a given proteome[70]. The Mascot score of a given peptide is equal to: −10 × Log₁₀(P), where P is the absolute probability. The Mascot scores for individual proteins were then calculated based upon the summation of the individual peptides for all peptides matching a given protein. The samples analyzed by LC-MS include purifications from control cell lines, Strep-METTL2A, Strep-METTL2B, and Strep-METTL6. The LC-MS analysis was performed once with each sample.

### In vitro methyltransferase assay.
tRNA templates were created by in vitro transcription. A list of single-stranded 4 nmol Ultramer DNA oligos (IDT) used as templates are listed in Supplementary Table 1. Each template was designed to harbor a T7 promoter upstream of the tRNA gene sequence. Each template was PCR amplified using a T7 Forward Primer (listed in Supplementary Table 1) and a specific reverse primer complementary to the 3′ end of each tRNA species. PCR amplification was done using Herculase II Fusion DNA Polymerase (Agilent Technologies) following standard procedures. The PCR parameters are as follows: 95 °C for 2 min followed by 35 cycles of 95 °C for 20 s, 48 °C for 20 s and 72 °C for 30 s, ending with 72 °C for 2 min. The PCR products were then resolved on a 2% agarose gel where bands were excised, and DNA was purified using the Qiagen Gel Extraction Kit. In vitro transcription was done using Optizyme T7 RNA Polymerase (ThermoFisher) following standard procedures. Reactions were incubated at 37 °C for 3 h followed by DNase treatment (RQ1 DNase, Promega) at 37 °C for 30 min. RNA was then purified using RNA Clean and Concentrator Zymo-Spin IC Columns (Zymo Research). tRNA transcripts were visualized on a 15% poly-acrylamide, 7 M urea gel stained with SYBR Gold nucleic acid stain.

Proteins to be used in methyltransferase assays were expressed in 293T cells by transient transfection and purified by above procedures. In all, 5 μl of input and purified protein was visualized by immunoblotting to ensure proper protein expression before use in methyltransferase assays. Prior to incubation with purified protein from human cells, the tRNA was first refolded by initial denaturation in 5 mM TRIS pH 7.5 and 0.16 mM EDTA and heated to 95 °C for two minutes before a two-minute incubation on ice. Refolding was conducted at 37 °C for 20 min in the presence of HEPES pH 7.5, MgCl₂, and NaCl.

For each methyltransferase reaction, 100 ng of refolded tRNA was incubated with 42 nM of purified protein elution along with 50 mM TRIS pH 7.5, 0.1 mM EDTA, 1 mM DTT, 0.5 mM S-adenosylmethionine (NEB) for 4 h at 30 °C. RNA was purified using RNA Clean and Concentrator Zymo-Spin IC Columns where the RNA was resuspended in 6 μl of RNase-free water. To observe if methylation at residue 32 occurred, we utilized primer extension analysis where reverse transcriptase extension would be impeded if methylation occurred. Extracted tRNA was pre-annealed with 0.625 pmol of a 5′-³²P-radiolabelled oligonucleotide that hybridizes around 6 nucleotides downstream residue 32 with 2.8 μl 5X hybridization buffer (250 mM TRIS pH 8.5, 300 mM NaCl) to a total of 14 μl. The mixture was heated at 95 °C for 3 minutes and then allowed to slowly cool to either 58 °C. In all, 14 μl of extension mix (0.12 μl of avian myeloblastosis virus reverse transcriptase (Promega), 2.8 μl 5X RT buffer, 1.12 μl 1 mM deoxynucleotidetriphosphates, RNase-free water to 14 μl) was added to each reaction and incubated at 58 °C for 1 h. Samples were then mixed with 2X formamide denaturing dye, heated to 95 °C for 3 min and resolved on an 18% polyacrylamide, 7 M urea denaturing gel. Gels were exposed by Phosphor-Imager analysis. Positive controls underwent same treatment but started with 3.2 μg of RNA.

### Immunoblot and Northern blot assays.
To verify loss of DALRD3, cell extracts were loaded onto BOLT 4–12% Bis-Tris gels (ThermoFisher) followed by immunoblotting onto PVDF membrane and probing with DALRD3 antibody (Proteintech, cat. No. 26294-1-AP, 1:1000 dilution) and anti-Actin C4 (EMD Millipore, cat. No MAB1501, 1:000 dilution). Expression of DALRD3 in stably-infected rescue cell lines were characterized by immunoblotting with the anti-FLAG M2 antibody (Sigma-Aldrich, F1804, 1:3333 dilution). Transient transfections of TWIN-Strep and FLAG-tagged proteins were also loaded onto BOLT 4–12% Bis-Tris gels, immunoblotted to PVDF membranes and probed with anti-TWIN-Strep (Genscript, cat. No. A01732, 1:1000 dilution) and anti-FLAG M2. NLuc immunoblots were probed with anti-FLAG, anti-GFP (Santa Cruz, cat. No. sc-9996; 1:100 dilution) and anti-Actin C4 antibodies. Image analysis of immunoblots were performed using Image Studio software (Li-Cor).

3-methylcytidine modification status was explored through the PHA assay. To conduct the PHA assay, probes were designed to hybridize upstream and downstream of residue 32 (oligos listed in Supplementary Table 1). 5 μg of RNA was loaded onto a 10% polyacrylamide, 1xTBE, 7 M urea gel and transferred onto an Amersham Hybond-XL membrane (GE Healthcare) for Northern Blotting analysis. Oligonucleotides used to detect RNAs are listed in Supplementary Table 1. The oligos were radiolabeled by T4 polynucleotide kinase (NEB) with adenosine

[γ32P]-triphosphate (6000 Ci/mmol, Amersham Biosciences) following standard procedures. Northern blots were visualized by Phosphor-Imager analysis and stripped via two incubations at 80 °C for 30 min in a buffer containing 0.15 M NaCl, 0.015 m Na-citrate and 0.1% SDS. Image analysis of Phosphorimager scans were performed using ImageJ open source software.

**Human subjects.** Evaluation of affected members by a board-certified clinical geneticist included obtaining medical and family histories, clinical examination, neuroimaging and clinical laboratory investigations. After obtaining a written informed consent for enrollment in an IRB-approved project from the Research Advisory Council (RAC) of the King Faisal Specialist Hospital and Research Centre (KFSHRC RAC# 2121053), venous blood was collected in EDTA and sodium heparin tubes for DNA extraction and establishment of lymphoblastoid cell lines (patients 19DG0509 and 19DG0510), respectively. All studies abide by the Declaration of Helsinki principles. The authors affirm that the parents of the human research participants provided informed consent for publication of the images of the minors in Fig. 6.

**Statistical analyses and reproducibility.** All statistics and graphs were performed and generated using GraphPad Prism software. Where applicable, error bars represent the standard deviation. Statistical tests and the number of times each experiment was repeated are stated in each figure legend. For primer extension assays, statistical analyses of the quantification were performed using one-way ANOVA and significance calculated using Tukey's multiple comparison test. For the codon reporter assays, statistical analyses for significance was calculated a two-tailed, unpaired students $t$-test ($t = 1.831$, df = 4).

**Reporting summary.** Further information on research design is available in the Nature Research Reporting Summary linked to this article.

## Data availability

The source data underlying all Figures are provided as a Source data file. The mass spectrometry data sets that include peptide spectral matches are included as Supplementary Data 1 and 2. All data is available from the corresponding author upon reasonable request.

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

## Acknowledgements

We thank Eric Phizicky, Sina Ghaemmaghami and members of the Fu Lab for comments on the paper; Kevin Welle and the URMC Mass Spectrometry Resource Lab for proteomics; and Jon Lueck for assistance in design of the codon reporter constructs. This work was supported by the Saudi Human Genome Program, King Salman Center for Disability Research, and King Abdulaziz City for Science and Technology Grant 08-MED497-20 to F.S.A. and National Science Foundation CAREER Award 1552126 to D.F.

## Author contributions

Conceptualization, J.L. and D.F.; methodology, J.L., F.S.A, and D.F.; investigation, J.L., H.S.A., E.F., F.S.A., and D.F.; writing—original draft, J.L and D.F.; writing—review and editing, J.L., E.F., F.S.A., and D.F.; funding acquisition, F.S.A., and D.F.; resources, E.F., F.S.A., and D.F.; supervision, E.F., F.S.A., and D.F.

## Competing interests

The authors declare no competing interests.
