## [Peer Review File · Nature Communications]

Reviewers' comments:

Reviewer #1 (Remarks to the Author):

The authors have demonstrated that METTL2A and 2B proteins interact with DALRD3, a protein of prior unknown function that has homology to Arg tRNA synthetase in the anticodon binding domain. They showed that DALRD3 promotes METTL2A/2B association of tRNAs-Arg(CCU/UCU) but not other tRNAsArg nor other isoacceptor tRNAs, and that a METTL2A/2B-DALRD3 complex isolated from cells has m3C32 activity on in vitro synthesized tRNAs-Arg(CCU/UCU). They showed that m3C32 formation specificity of the METTL2A/2B-DALRD3 complex resides in the U36-A37 sequence specific to the tRNAs-Arg(CCU/UCU) but not other tRNAsArg isoacceptors. This makes a significant advance that nicely accounts for the presence of m3C32 on tRNAs-Arg(CCU/UCU) in vertebrates but not other tRNAsArg whereas in yeasts the tRNAs-Ser and Thr also contain m3C32. This adds to our understanding of an evolving family of RNA methyltransferases of the Trm140 type, some of which are known to use cofactors that may recognize 'identity' elements in their specific substrates. In a significant 'human genetics' extension of this, the authors identified two siblings with homozygous nonsense mutations in DALRD3 with a severe neurodevelopmental disorder and demonstrable hypomodification of m3C32 in ArgCCU/UCU. Most of the most important data support the main conclusions. There are scientific weaknesses and other issues that should be addressed as follows.

Science Data issues:

1.) p.7 bottom-8 top, referring to Fig 2C "Notably, we found that DALRD3 purifications were greatly enriched for arginine tRNAs containing m3C generated by METTL2A/B (Figure 2C.." The problem with this sentence is that the authors do not know if the associated arginine tRNAs contain m3C32 because they have not assayed those tRNAs for the modification. It might be that the stably associated tRNAs are the substrates that have not yet been modified. They should change the sentence to "enriched for those tRNAs known to contain m3C".

Directly related to this, the right panel of and figure legend for Fig 3C are misrepresented/mislabeled. The legend states "the presence or absence of m3C" but the authors have not assayed for the presence or absence of m3C32. Again, the co-purified tRNAs may be unmodified substrates.

The authors can examine/address this issue in Fig 2C and 3C by use of the PHA assay on those northern blots; the % m3C32 modified of the co-purified tRNA-Arg-CCU/UCU vs. the % modified of total cellular tRNA-Arg-CCU/UCU will reveal if the associated RNAs are hypomodified or not.

Another way to address this aspect of tRNA substrate binding may be to perform the expts in fig 2c & 3c with catalytically inactive METTL2A/B mutant and ask if the % yield changes relative to WT METTL2A/B.

2) Fig 5F-H, trying to map the domains of DALRD3 is the weakest part of the data, is unconvincing as is, and detracts from the overall impact. The data are too weak to conclude that delta-N-term is more active than delta-DALR or vector (background); an internal calibration is needed, a ratio of m3C to D20 bands in all lanes.

3) Fig 6F should include control tRNAs Arg-ACG, Ser, Thr, as was done for previous figures.

4) For Figs 2A and 3A, ideally there should be loading controls.

5) Fig 5A CRISPR KO no mention of the asterisk next to the stronger band in both lanes. This data panel is not convincing as is, -however, the PHA assay provides strong functional evidence of the DALRD3 KO. Similar but worse problem for Fig 6E, poor quality WB. The authors should probably note that the patient DALRD3 mRNA may be subjected to NMD, -have they tried to detect mRNA levels from lymphoblastoid cell lines (LCLs)?

6) "Parents are healthy first cousins and they have one healthy child in addition to the two affected children (Figure 6B)." However, the family tree shows 4 offspring with no key/details in legend. this should be fixed.

For description fig 6, it says "identified a consanguineous family" Authors should describe how they identified the patients/family.

- For Fig 4C, D; could have used METTL2A/B purified w/o DALRD3 as a control.

- Can the authors comment on if the interactions between the METTL2 proteins and DALRD3 proteins in the complexes are sensitive to ribonuclease?

Other issues to be addressed:

A) Introduction p3, "S. pombe expresses two Trm140 homologs encoded by the Trm140 and Trm141 genes that are separately responsible for catalyzing m³C in tRNA-Ser and tRNA-Thr, respectively" is incorrect. Trm140 modifies tRNA-Thr and Trm141 modifies tRNA-Ser, and is again incorrect referring to Trm141 in middle p. 4.

B) p.12 top. although earlier citations are of data from microarrays and crystal structures (Hiley et al, Pallan et al), the PHA assay as a Northern blot approach that uses a second probe as an internal control to gauge relative hypomodification of specific tRNA, was first reported by Lamichhane et al 2011 RNA 17:1846-57, and should be cited appropriately.

Discussion:

The authors should consider that METTL2A/B are the sequence specific recognition proteins and DALRD3 activates their modification activity.

With regard to association of neurological disorders and hypomodification of mitochondrial tRNA, should note that DALRD3 dependency of Arg m³C₃₂ formation is limited to cytosolic tRNAs, i.e., there are no known mito-tRNA substrates. However, they should also note the possibility of non-tRNA substrates.

With regard to the "outstanding question" raised in Discussion on p.16, the spectrum of DALRD3-associated RNAs identified in this study is limited, by specific probing of candidate tRNAs, -should do

more broad exam including by a -Seq approach.

With regard to the last paragraph on p.16 of Discussion, this might be a good place to note the presence of vertebrate-specific m³C47:3 in tRNA^{Ser}.

Reviewer #2 (Remarks to the Author):

In this study, Lentini et al. investigate the modification of a certain arginine tRNA isoacceptors by the methyltransferase METTL2, and discover that this enzyme interacts with a novel protein, DALRD3, that underlies the specificity of METLL2 to the UCU and CCU families of arginine tRNA. The evidence presented for a functional interaction between METLL2 and DALRD3 is strong and convincing, and the genetic evidence linking a mutation in DALRD3 to neurological disease underscores the importance of understanding the molecular mechanism of tRNA recognition by METTL2.

Points to address:

1. When cotransfected, the authors found that overexpression of DALRD3 resulted in reduced levels of METTL2. Does loss of endogenous DALRD3 expression, as in the HAP1 KO line affect the expression/stability of METLL2?
2. While the authors indicate in their figure legends that most experiments have been repeated 3 times, for the most part, they do not show any quantification or statistics. While the primer extension assay in figure 4D, has been quantified, there is still no statistical analysis.
3. Both the western blots and the northern blots throughout the manuscript lack any sort of molecular weight marker.
4. In Figure 1B, the authors include the mascot score of their proteins, however, they do not explain this score in the figure legend nor in the methods. An explanation needs to be added, or this score should be changed to a more easily understood statistical measurement.
5. Could the authors add residue numbers to the anticodon loops shown in Figure 4A and B?
6. Could the authors show a higher exposure for the IPs in Figure 3A? Its extremely difficult to see the bands in lane 11.
7. In figure 5H, its difficult to distinguish the possible m³C band formed upon expression of the N-terminal deletion construct. If the authors believe that expression of this construct does indeed result in

a slight increase in m³C formation—quantification is necessary.

8. In the materials and methods, the northern protocol is actually under ‘transient transfections, protein purifications and RNA purifications’ and not in the ‘Immunoblot and northern blot assays’ section.

9. An explanation of the rings (homozygous regions vs SNPS etc) on the multideogram would be helpful in the legend for Fig 6.

Reviewer #3 (Remarks to the Author):

This work describes the identification of the DALRD3 protein as a co-factor for arg tRNA m³C modification. Human tRNAs are modified at many places and with many types. M³C is present in the tRNAs for Ser, Thr, and Arg. Although the methyltransferase enzymes for these tRNAs are previously known, how m³C is installed specifically in Arg tRNA isoacceptors with CCU/UCU anticodon was not known. Lentini et al first performed biochemical protein association studies to find DALRD3 as a strongly associated protein to the Arg tRNA methyltransferase, METTL2A/B. They then show that this complex is also present in cells, and DALRD3 is necessary to confer the m³C modification specificity for METTL2. Additional in vitro studies identified the arg tRNA sequence identity that distinguishes the CCU/UCU isoacceptors from other arg tRNAs. Furthermore, the authors found a pathological human DALRD3 gene mutation that led to a truncated DALRD3 protein and the loss of arg tRNA m³C modification. Although eukaryotic tRNA modifications have been extensively studied in yeast, our understanding of human tRNA modifications still have a lot of gaps. This well-designed and well controlled study expands our knowledge on human tRNA modifications with an additional emphasis on the relationship to a human disease.

1. (major) The authors need to provide some results on how this specific modification in arg tRNA affects arg tRNA-dependent cellular processes. The most obvious, potential function of m³C³² arg tRNA modification is to affect global translation in an Arg-codon dependent manner. CCU/UCU tRNAs specifically decode AGG/AGA codons. There is a precedent of a specific arg tRNA isodecoder with UCU anticodon that alleviates ribosome stalling at AGA codons, revealed by ribosome profiling (Science. 2014 Jul 25;345(6195):455-9). The authors should perform a similar experiment with their WT and DALRD3-KO cell lines to test this hypothesis.

2. (major) Another test of the decoding hypothesis is to insert a run of AGA or AGG codons in a reporter system, e.g. the dual luciferase, as is commonly done in the field.

3. (minor) p.11, last paragraph: the DNA sequencing result of the Crispr/Cas generated D3-KO line should be shown as a supplemental figure.

RESPONSE TO REVIEWERS

We thank all reviewers for their insight and suggestions that have improved the quality and rigor of this manuscript. Through the requested edits and experiments, we have provided additional results and quantification that further support and extend our discoveries. Please find below a detailed response to each Reviewer point.

Reviewers' comments:

Reviewer #1 (Remarks to the Author):

The authors have demonstrated that METTL2A and 2B proteins interact with DALRD3, a protein of prior unknown function that has homology to Arg tRNA synthetase in the anticodon binding domain. They showed that DALRD3 promotes METTL2A/2B association of tRNAs-Arg(CCU/UCU) but not other tRNAsArg nor other isoacceptor tRNAs, and that a METTL2A/2B-DALRD3 complex isolated from cells has m3C32 activity on in vitro synthesized tRNAs-Arg(CCU/UCU). They showed that m3C32 formation specificity of the METTL2A/2B-DALRD3 complex resides in the U36-A37 sequence specific to the tRNAs-Arg(CCU/UCU) but not other tRNAsArg isoacceptors. This makes a significant advance that nicely accounts for the presence of m3C32 on tRNAs-Arg(CCU/UCU) in vertebrates but not other tRNAsArg whereas in yeasts the tRNAs-Ser and Thr also contain m3C32. This adds to our understanding of an evolving family of RNA methyltransferases of the Trm140 type, some of which are known to use cofactors that may recognize 'identity' elements in their specific substrates. In a significant 'human genetics' extension of this, the authors identified two siblings with homozygous nonsense mutations in DALRD3 with a severe neurodevelopmental disorder and demonstrable hypomodification of m3C32 in ArgCCU/UCU. Most of the most important data support the main conclusions. There are scientific weaknesses and other issues that should be addressed as follows.

RESPONSE: We thank the reviewer for their comments and suggestions that have improved the clarity and scientific rigor of our manuscript.

Science Data issues:

1.) p.7 bottom-8 top, referring to Fig 2C "Notably, we found that DALRD3 purifications were greatly enriched for arginine tRNAs containing m3C generated by METTL2A/B (Figure 2C.." The problem with this sentence is that the authors do not know if the associated arginine tRNAs contain m3C32 because they have not assayed those tRNAs for the modification. It might be that the stably associated tRNAs are the substrates that have not yet been modified. They should change the sentence to "enriched for those tRNAs known to contain m3C".

RESPONSE: We have edited the sentence as suggested. The sentence now reads "*Notably, we found that DALRD3 purifications were greatly enriched for arginine tRNAs known to contain m3C generated by METTL2A/B (Fig. 2c, lanes 10-12, tRNA-Arg-CCU and UCU).*"

Directly related to this, the right panel of and figure legend for Fig 3C are misrepresented/mislabeled. The legend states "the presence or absence of m3C" but the authors have not assayed for the presence or absence of m3C32. Again, the co-purified tRNAs may be unmodified substrates.

RESPONSE: We concur with the Reviewer that the panel and legend could be misinterpreted as demonstrating the presence of m3C in the co-purifying tRNAs. The intention of the right panel was to denote which tRNAs have been previously characterized to contain m3C. To provide further clarification, we have revised the Figure legend to the following: "*The known presence or absence of m3C and the METTL enzyme that generates m3C in a given tRNA is denoted on the right*".

We thank the Reviewer improving the clarity of our manuscript.

The authors can examine/address this issue in Fig 2C and 3C by use of the PHA assay on those northern blots; the % m3C32 modified of the co-purified tRNA-Arg-CCU/UCU vs. the % modified of total cellular tRNA-Arg-CCU/UCU will reveal if the associated RNAs are hypomodified or not.

Another way to address this aspect of tRNA substrate binding may be to perform the expts in fig 2c & 3c with catalytically inactive METTL2A/B mutant and ask if the % yield changes relative to WT METTL2A/B.

RESPONSE: We have followed the Reviewer's suggestion and probed the Northern blots of endogenous and co-purified tRNAs with PHA probes (now included as Supplemental Figure 3).

We find that DALRD3 co-purifies with tRNA-Arg-CCU or Arg-UCU that exhibits strong PHA signal indicative of the unmodified form of tRNA-Arg-CCU and UCU (Supplemental Figure 3a, tRNA-Arg-CCU and UCU, PHA probes, lanes 10-12). In contrast, METTL2A and METTL2B interact with tRNA-CCU or Arg-UCU that display weak PHA signal indicative of the modified form (Supplemental Figure 3b, tRNA-Arg-CCU and UCU, PHA probes, lanes 11 and 12). These results are consistent with DALRD3 binding unmodified tRNA-Arg-UCU and CCU for subsequent methylation upon formation of the METTL2-DALRD3 complex.

We thank the Reviewer for the experimental suggestion that has provided further insight into our findings.

2) Fig 5F-H, trying to map the domains of DALRD3 is the weakest part of the data, is unconvincing as is, and detracts from the overall impact. The data are too weak to conclude that delta-N-term is more active than delta-DALR or vector (background); an internal calibration is needed, a ratio of m3C to D20 bands in all lanes.

RESPONSE: We agree with the Reviewer that our previous studies on the DALRD3 domains lacked conclusive evidence. Thus, we have followed the Reviewer's suggestion and have repeated the experiments in Figure 5F-H to perform more precise quantification using an internal calibration. The new results have been added to Figure 5 as panels "h" and "i".

Based upon these results, we find that neither the DALRD3 Δ N-term nor Δ DALR variant is able to rescue m³C modification in the DALRD3-KO cell line to any appreciable extent. Furthermore, we find no difference in the ability to rescue m³C formation between the DALRD3 Δ N-term and Δ DALR variants. We thank the Reviewer for their suggestion that has strengthened our conclusions.

3) Fig 6F should include control tRNAs Arg-ACG, Ser, Thr, as was done for previous figures.

RESPONSE: We have added the PHA assays for tRNA-Ser and Thr as panel “b” in Figure 7. As expected, we find no difference in PHA signal for tRNA-Ser-UGA or Thr-AGU between WT and patient LCLs.

4) For Figs 2A and 3A, ideally there should be loading controls.

RESPONSE: We have added a loading control for the immunoblots shown in Figures 2 and 3. The loading control is a non-specific band that is detected by the anti-Strep tag antibody. The non-specific band can be detected in all the input lanes but is not present in the purified samples.

5) Fig 5A CRISPR KO no mention of the asterisk next to the stronger band in both lanes. This data panel is not convincing as is, -however, the PHA assay provides strong functional evidence of the DALRD3 KO.

RESPONSE: We apologize for neglecting to define the asterisk in Figure 5A. The asterisk denotes a non-specific band that was detected using the anti-DALRD3 antibody. We have revised the Figure legend to define the asterisk.

Similar but worse problem for Fig 6E, poor quality WB. The authors should probably note that the patient DALRD3 mRNA may be subjected to NMD, -have they tried to detect mRNA levels from lymphoblastoid cell lines (LCLs)?

RESPONSE: We agree with the Reviewer’s comment that the immunoblot in Figure 6E is far from ideal due to multiple non-specific bands. After testing several anti-DALRD3 antibodies from commercial vendors, we found only one antibody that was able to detect DALRD3 in human whole cell lysates based upon the molecular weight of DALRD3 and its absence in DALRD3-deficient cells (see Figure 5a, Proteintech, cat. No. 26294-1-AP). Unfortunately, even after optimization, this antibody detects non-specific proteins with much higher intensity than DALRD3. Thus, we are limited by the weak specificity and sensitivity of commercially-available antibodies against DALRD3. However, as noted by the Reviewer, our functional assays provide independent evidence that DALRD3 activity has been abolished in the patient cell lines.

The authors should probably note that the patient DALRD3 mRNA may be subjected to NMD, -have they tried to detect mRNA levels from lymphoblastoid cell lines (LCLs)?

RESPONSE: We agree with the Reviewer that the *DALRD3* variant mRNA could be subject to NMD but have not verified this possibility in the patient LCLs. To note this potential outcome, we have added the following text: “*In addition, the variant DALRD3 mRNA transcript could be subject to nonsense mediated decay due to the presence of a premature stop codon.*”

6) "Parents are healthy first cousins and they have one healthy child in addition to the two affected children (Figure 6B)." However, the family tree shows 4 offspring with no key/details in legend. this should be fixed.

RESPONSE: We apologize for the confusion caused by the pedigree symbol for a spontaneous miscarriage that can be easily misunderstood as an “offspring”. We have revised the sentence to read “*Parents are healthy first cousins and they have one healthy child in addition to the two affected children and one spontaneous miscarriage*”.

For description fig 6, it says "identified a consanguineous family" Authors should describe how they identified the patients/family.

RESPONSE: The family was identified by applying clinical exome sequencing as a first-tier test for newly enrolled patients with neurodevelopmental disorders. To clarify how we identified the family, we have added the sentence: “*The family was identified by applying a “genomics first” approach to uncharacterized patients with neurodevelopmental disorders as previously described⁴⁹.*”

•For Fig 4C, D; could have used METTL2A/B purified w/o DALRD3 as a control.

RESPONSE: As suggested, we have performed the *in vitro* assays using METTL2A/B purified with or without co-expression of DALRD3. In this case, we can still detect methyltransferase activity for purified METTL2 due to the co-purification of endogenous DALRD3. This is not surprising since this is how we found DALRD3 in the first place via the purification of METTL2 and the detection of endogenous DALRD3 by mass spectrometry. These results have been included as panel c in Figure 4.

•Can the authors comment on if the interactions between the METTL2 proteins and DALRD3 proteins in the complexes are sensitive to ribonuclease?

RESPONSE: We have followed the Reviewer’s suggestion and tested whether the METTL2-DALRD3 complex is sensitive to RNase treatment. Using this approach, we find that METTL2 retains interaction with DALRD3 after RNase degradation of the co-purifying tRNA. These results suggest that the interaction between METTL2 and DALRD3 is not dependent upon a bridging tRNA molecule. The new results have been included as Supplemental Figure 2.

Other issues to be addressed:

A) Introduction p3, “*S. pombe* expresses two Trm140 homologs encoded by the Trm140 and Trm141 genes that are separately responsible for catalyzing m³C in tRNA-Ser and tRNA-Thr, respectively” is incorrect. Trm140 modifies tRNA-Thr and Trm141 modifies tRNA-Ser, and is again incorrect referring to Trm141 in middle p. 4.

RESPONSE: We have made the aforementioned changes to the text and thank the reviewer for the careful reading of our manuscript.

B) p.12 top. although earlier citations are of data from microarrays and crystal structures (Hiley et al, Pallan et al), the PHA assay as a Northern blot approach that uses a second probe as an internal control to gauge relative hypomodification of specific tRNA, was first reported by Lamichhane et al 2011 RNA 17:1846-57, and should be cited appropriately.

RESPONSE: We apologize for the omission and have corrected the text to include this citation.

Discussion:

The authors should consider that METTL2A/B are the sequence specific recognition proteins and DALRD3 activates their modification activity.

RESPONSE: We agree with the Reviewer that METTL2A/B could be the main source of sequence specificity while DALRD3 activates METTL2A/B activity. Thus, we have revised the text to reflect this potential mechanism for tRNA substrate recognition in m³C formation. We have added this possibility in the Discussion section as: *“Another possibility is that METTL2 contributes primarily to substrate specificity while DALRD3 serves as an activator of METTL2 methyltransferase activity.”*

With regard to association of neurological disorders and hypomodification of mitochondrial tRNA, should note that DALRD3 dependency of Arg m³C₃₂ formation is limited to cytosolic tRNAs, i.e., there are no known mito-tRNA substrates.

RESPONSE: We have added a note in the Discussion section that DALRD3 dependency of m³C₃₂ formation is limited to cytosolic tRNA since there are no known mitochondrial tRNA substrates.

The revised sentence now reads: *“While there are no known mitochondrial tRNAs dependent upon DALRD3 for m³C formation, there could be a possible indirect impact of DALRD3-dependent tRNA modification on mitochondrial homeostasis since cytoplasmic tRNA modifications in the anticodon loop have been shown to impact mitochondrial metabolism⁴³.”*

However, they should also note the possibility of non-tRNA substrates.

With regard to the "outstanding question" raised in Discussion on p.16, the spectrum of DALRD3-associated RNAs identified in this study is limited, by specific probing of candidate tRNAs, -should do more broad exam including by a -Seq approach.

RESPONSE: Indeed, it would be fascinating if DALRD3 associates with additional RNAs besides tRNAs to target them for modification. While outside the scope of the current manuscript, we agree with the Reviewer that a broad examination of DALRD3-associated RNAs will be an interesting experiment to perform in the future.

We have updated the text in the Discussion to include the possibility of non-tRNA substrates: “For example, DALRD3 may interact with additional RNAs besides tRNAs to modulate their modification status or stability.”

With regard to the last paragraph on p.16 of Discussion, this might be a good place to note the presence of vertebrate-specific m³C_{47:3} in tRNA^{Ser}.

RESPONSE: We have added this note on m³C-47:3 in tRNA-Ser isoacceptors to our Discussion section. The added sentence in the Discussion reads: “We also note that m³C at position 47:3 of the variable arm is present in mammalian tRNA-Ser isoacceptors but not *S. cerevisiae* tRNAs, suggesting the possibility of another mammalian-specific tRNA methyltransferase complex that remains to be identified.”

Reviewer #2 (Remarks to the Author):

In this study, Lentini et al. investigate the modification of a certain arginine tRNA isoacceptors by the methyltransferase METTL2, and discover that this enzyme interacts with a novel protein, DALRD3, that underlies the specificity of METLL2 to the UCU and CCU families of arginine tRNA. The evidence presented for a functional interaction between METLL2 and DALRD3 is strong and convincing, and the genetic evidence linking a mutation in DALRD3 to neurological disease underscores the importance of understanding the molecular mechanism of tRNA recognition by METTL2.

RESPONSE: We thank the reviewer for their positive comments and suggestions that have improved our manuscript.

Points to address:

1. When cotransfected, the authors found that overexpression of DALRD3 resulted in reduced levels of METTL2. Does loss of endogenous DALRD3 expression, as in the HAP1 KO line affect the expression/stability of METLL2?

RESPONSE: This is an excellent question that we have resolved by comparing the levels of METTL2A between the WT and DALRD3-KO cell lines. Based upon immunoblotting, we find no detectable change in METTL2A levels in the DALRD3-KO cell line. Thus, it appears that METTL2A expression/stability is not majorly impacted by DALRD3-deficiency. The new data has been included as Supplemental Figure 5, panel “b”.

2. While the authors indicate in their figure legends that most experiments have been repeated 3 times, for the most part, they do not show any quantification or statistics. While the primer extension assay in figure 4D, has been quantified, there is still no statistical analysis.

RESPONSE: We have followed the Reviewer’s request and provided further quantification and statistics in the manuscript, including the primer extension assays in Figures 4, 5 and 7. The figures with quantification now include:

Figure 1a. MASCOT score for peptide matching confidence
Figure 1c. % DALRD3 purified via immunoblot
Figure 2c. % yield for each copurifying RNA
Figure 4e. Primer extension quantification and statistical analysis
Figure 5i. Primer extension quantification and statistical analysis
Figure 7d. Primer extension quantification and statistical analysis

We thank the Reviewer for improving the rigor of our manuscript.

3. Both the western blots and the northern blots throughout the manuscript lack any sort of molecular weight marker.

RESPONSE: We apologize for the omission of molecular weight markers which will be important for data interpretation. Thus, we have included the full immunoblots in our Source Data file, which shows the labeled molecular weight markers for all gels shown in this manuscript. Moreover, we have labeled the molecular weight markers for the immunoblots in each Figure. We thank the Reviewer for pointing out this omission.

4. In Figure 1B, the authors include the mascot score of their proteins, however, they do not explain this score in the figure legend nor in the methods. An explanation needs to be added, or this score should be changed to a more easily understood statistical measurement.

RESPONSE: We have added an explanation for the Mascot score to the Methods section. The Methods section now reads:

The Mascot scores of individual peptides were calculated as the absolute probability that the observed peptide match is a random event when matching spectra to all the expected spectra of a given proteome (Koenig et al 2008). The Mascot score of a given peptide is equal to: $-10 \times \text{Log}_{10}(P)$, where P is the absolute probability. The Mascot scores for individual proteins were then calculated based upon the summation of the individual peptides for all peptides matching a given protein.

We thank the Reviewer for pointing out this missing information.

5. Could the authors add residue numbers to the anticodon loops shown in Figure 4A and B?

RESPONSE: We have followed the Reviewer's suggestion and numbered nucleotide positions 32 and 36 of the anticodon loop. The labeling of positions 32 and 36 will allow the reader to count backward or forward from these nucleotides to locate a specific nucleotide. For space and clarity reasons, we did not label the entire anticodon loop since it could cause confusion with the numbering of the chemical modifications such as mcm5s2U. We thank the Reviewer for the suggestion.

6. Could the authors show a higher exposure for the IPs in Figure 3A? Its extremely difficult to see the bands in lane 11.

RESPONSE: We have increased the exposure for IPs in Figure 3A. The bands in lane 11 are now readily visible. We thank the Reviewer for the suggestion.

7. In figure 5H, its difficult to distinguish the possible m3C band formed upon expression of the N-terminal deletion construct. If the authors believe that expression of this construct does indeed result in a slight increase in m3C formation—quantification is necessary.

RESPONSE: We have followed the Reviewer's request and repeated the experiments in Figure 5F-H in order to perform quantification. The new results and quantification have been added to

Figure 5 as panels h and i. Based upon these results, we find that neither the DALRD3 Δ N-term nor Δ DALR variant is able to rescue m3C modification in the DALRD3-KO cell line to any appreciable extent. Furthermore, we find no difference in the ability to rescue between the DALRD3 Δ N-term and Δ DALR variants. We thank the Reviewer for the critique that has increased the rigor of our findings.

8. In the materials and methods, the northern protocol is actually under ‘transient transfections, protein purifications and RNA purifications’ and not in the ‘Immunoblot and northern blot assays’ section.

RESPONSE: This information has moved to the proper location in the materials and methods. We thank the reviewer for noticing this error.

9. An explanation of the rings (homozygous regions vs SNPS etc) on the multideogram would be helpful in the legend for Fig 6.

RESPONSE: We have added text to the legend of Figure 6. The legend now reads:

AgileMultiIdeogram output of autozygosity analysis of the study family as a series of block arcs with the two affected individuals represented by the outer two arcs. Autozygous regions from affected individuals are marked as pale blue, those from unaffected individuals as pink, and those homozygous in all affected individuals as dark blue.

Reviewer #3 (Remarks to the Author):

This work describes the identification of the DALRD3 protein as a co-factor for arg tRNA m³C modification. Human tRNAs are modified at many places and with many types. M³C is present in the tRNAs for Ser, Thr, and Arg. Although the methyltransferase enzymes for these tRNAs are previously known, how m³C is installed specifically in Arg tRNA isoacceptors with CCU/UCU anticodon was not known. Lentini et al first performed biochemical protein association studies to find DALRD3 as a strongly associated protein to the Arg tRNA methyltransferase, METTL2A/B. They then show that this complex is also present in cells, and DALRD3 is necessary to confer the m³C modification specificity for METTL2. Additional in vitro studies identified the arg tRNA sequence identity that distinguishes the CCU/UCU isoacceptors from other arg tRNAs. Furthermore, the authors found a pathological human DALRD3 gene mutation that led to a truncated DALRD3 protein and the loss of arg tRNA m³C modification. Although eukaryotic tRNA modifications have been extensively studied in yeast, our understanding of human tRNA modifications still have a lot of gaps. This well-designed and well controlled study expands our knowledge on human tRNA modifications with an additional emphasis on the relationship to a human disease.

RESPONSE: We thank the Reviewer for their positive comments on our study and its significance.

1. (major) The authors need to provide some results on how this specific modification in arg tRNA affects arg tRNA-dependent cellular processes. The most obvious, potential function of m³C₃₂ arg tRNA modification is to affect global translation in an Arg-codon dependent manner. CCU/UCU tRNAs specifically decode AGG/AGA codons. There is a precedent of a specific arg tRNA isodecoder with UCU anticodon that alleviates ribosome stalling at AGA codons, revealed by ribosome profiling (Science. 2014 Jul 25;345(6195):455-9). The authors should perform a similar experiment with their WT and DALRD3-KO cell lines to test this hypothesis.

RESPONSE: We agree that it would be interesting to test how the m³C modification impacts translation. Indeed, we have followed the Reviewer's point #2 below using codon specific reporters. However, the ribosome profiling approach represents a major research study that would encompass an entire research manuscript by itself. Since this manuscript addresses the molecular mechanism by which an RNA modification enzyme identifies its target, the proposed ribosome profiling experiment lies well beyond the scope of this manuscript and would be more appropriate for a future study. We do appreciate the Reviewer's insightful suggestion and have included ribosome profiling as a future experiment in the Discussion section.

2. (major) Another test of the decoding hypothesis is to insert a run of AGA or AGG codons in a reporter system, e.g. the dual luciferase, as is commonly done in the field.

RESPONSE: We have followed the Reviewer's suggestion by generating and testing reporter constructs expressing Nano-luciferase fused to a run of arginine codons decoded by arginine tRNAs containing m³C. For this experiment, we used stable lentiviral integration of the codon

reporter due to the extremely low transfection efficiency of HAP1 haploid human cells (our unpublished observations, Findlay et al. Nature 2014, Alexander et al. BMC Biology 2019). The lentiviral expression constructs were designed similarly to a previously published translation reporter that employed stable integration (Darnell, Subramaniam and O'Shea EK. Mol Cell 2018). Using this system, we find that cells deficient in DALRD3 exhibited no major change in the expression of reporter proteins fused to either AGA or AGG codons, both of which are decoded by arginine tRNAs containing m3C. We have included these results as Supplementary Figure 6 in our revised manuscript.

These results suggest that the m3C modification has no observable effects on translation efficiency in this particular codon reporter system. Another possibility is that m3C modification plays a role in arginine tRNAs that cannot be detected using this experimental system. In the revised Discussion section, we provide potential reasons underlying this result, including the limitation that the arginine codon reporter system represents a highly specific context that might not reflect the actual physiological impact of the m3C modification on tRNA function in translation. While outside the scope of the current study, we have included the Reviewer's idea of ribosome profiling in the Discussion section as a possible future experiment to study the downstream functions of m3C modification in arginine tRNAs. We thank the Reviewer for their suggestion that has allowed us to further probe the role of m3C.

3. (minor) p.11, last paragraph: the DNA sequencing result of the Crispr/Cas generated D3-KO line should be shown as a supplemental figure.

RESPONSE: We have included the DNA sequencing results of the DALRD3-KO cell line in Supplemental Figure 5. We thank the Reviewer for improving the scientific rigor of the manuscript.

REVIEWERS' COMMENTS:

Reviewer #1 (Remarks to the Author):

After reading the authors' responses to the critiques and reading the revised manuscript, I feel that the points raised in the previous round of review have been very well addressed. The authors revised text and added significant new data in multiple figures that strengthened the manuscript including by providing insight to mechanism. The exception is in the case of reviewer #3, who noted as his/her #1 (major) point "The authors need to provide some results on how this specific modification in arg tRNA affects arg tRNA-dependent cellular processes.." The reviewer wrote that the authors should perform ribosome profiling in their WT and DALRD3-KO cell lines to determine if ribosome stalling is affected at AGA codons. I do not agree that there should be a "need" for such a large addition to this manuscript; I agree with the assessment the authors in their response to this point. The manuscript as is, in my opinion is a clear advance and does not require ribosome profiling nor the results of a reporter assay.

Reviewer #2 (Remarks to the Author):

The authors' revisions adequately address my concerns and I think the manuscript is now suitable for publication.

Reviewer #3 (Remarks to the Author):

The authors adequately addressed my comments.